# CoT3DReF: Chain-of-Thoughts Data-Efficient 3D Visual Grounding

**Eslam Mohamed Bakr, Mohamed Ayman, Mahmoud Ahmed, Habib Slim, Mohamed Elhoseiny**
King Abdullah University of Science and Technology (KAUST)
{eslam.abdelrahman, mohamed.mohamed.2, mahmoud.ahmed,
habib.slim, mohamed.elhoseiny}@kaust.edu.sa

## ABSTRACT

3D visual grounding is the ability to localize objects in 3D scenes conditioned by utterances. Most existing methods devote the referring head to localize the referred object directly, causing failure in complex scenarios. In addition, it does not illustrate how and why the network reaches the final decision. In this paper, we address this question *"Can we design an interpretable 3D visual grounding framework that has the potential to mimic the human perception system?"*. To this end, we formulate the 3D visual grounding problem as a sequence-to-sequence (Seq2Seq) task by first predicting a chain of anchors and then the final target. Interpretability not only improves the overall performance but also helps us identify failure cases. Following the chain of thoughts approach enables us to decompose the referring task into interpretable intermediate steps, boosting the performance and making our framework extremely data-efficient. Moreover, our proposed framework can be easily integrated into any existing architecture. We validate our approach through comprehensive experiments on the Nr3D, Sr3D, and Scanrefer benchmarks and show consistent performance gains compared to existing methods without requiring manually annotated data. Furthermore, our proposed framework, dubbed CoT3DRef, is significantly data-efficient, whereas on the Sr3D dataset, when trained only on 10% of the data, we match the SOTA performance that trained on the entire data. The code is available at *github.com/eslambakr/CoT3DVG*.

## 1 INTRODUCTION

The 3D visual grounding task involves identifying and localizing objects in a 3D scene based on a natural language description or query. This task is crucial for many applications, such as robotics (Nguyen et al., 2019; Karnan et al., 2022; Wijmans et al., 2019), virtual reality (Puig et al., 2018; Ghasemi et al., 2022; Park et al., 2020; Osborne-Crowley, 2020; Liu et al., 2019), and autonomous driving (Qian et al., 2022; Cui et al., 2021; Jang et al., 2017; Deng et al., 2021). The goal is to enable machines to understand natural language and interpret it in the context of a 3D environment. Although 3D visual grounding has significantly advanced, current solutions cannot imitate the human perception system nor be interpretable. To address this gap, we propose a Chain-of-Thoughts 3D visual grounding framework, termed CoT3DRef. One of the biggest challenges in machine learning is understanding how the model arrives at its decisions. Thus, the concept of Chain-of-Thoughts (CoT)

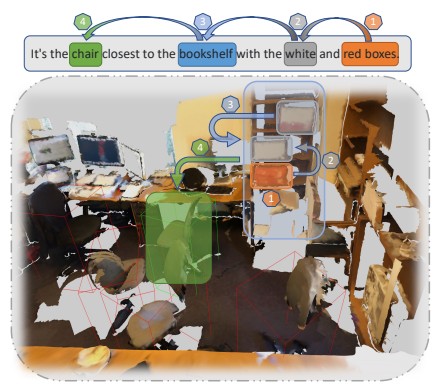

Figure 1: Overview of our approach, where we first predict a chain of anchors in a logical order. In this example, to reach the `chair` target, we first have to localize the `white and red boxes`, then the `bookshelf`.

comes in. Although CoT is widely applied in Natural Language Processing (NLP) applications (Wei et al., 2022; Chowdhery et al., 2022; Lyu et al., 2023; Wang et al., 2022; Zhang et al., 2023; Madaan & Yazdanbakhsh, 2022), it is less explored in vision applications. Understanding the CoT is crucial for several reasons. Firstly, it helps explain how the model arrived at its decision, which is essential for transparency and interpretability. Secondly, it helps identify potential biases or errors in the model,

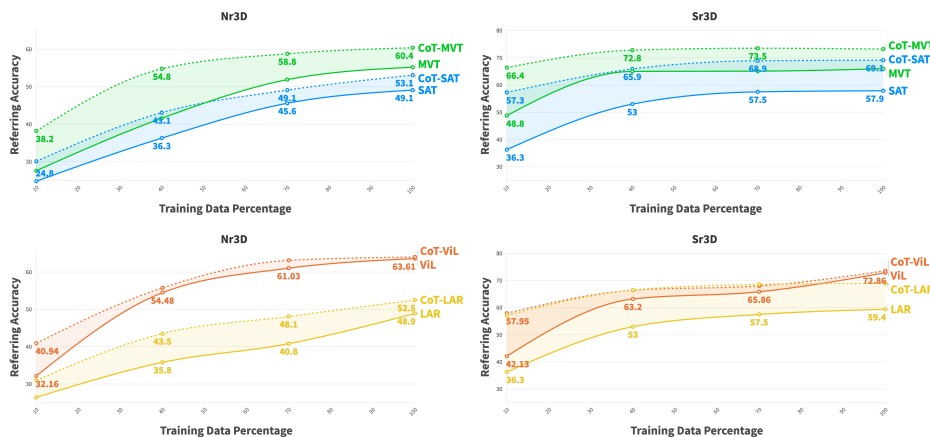

Figure 2: Data efficiency results. To show the effectiveness of our CoT architecture, we integrate it into four architectures, i.e., MVT (Huang et al., 2022), SAT (Yang et al., 2021), LAR (Bakr et al., 2022) and ViL (Chen et al., 2022), across different amounts of training data (10% - 100%).

which can be addressed to improve its accuracy and reliability. Third, it is a critical step toward intelligent systems that mimic human perception. Similar to machine learning models, our perception system can be thought of as a Chain-of-Thoughts (McVay & Kane, 2009; Chen et al., 2017) - a series of intermediate steps that enable us to arrive at our final perception of the world.

In this paper, we mainly answer the following question: Can we design an interpretable 3D visual grounding framework that has the potential to mimic the human perception system? To this end, we formulate the 3D visual grounding problem as a sequence-to-sequence (Seq2seq) task. The input sequence combines 3D objects from the input scene and an input utterance describing a specific object. On the output side, in contrast to the existing 3D visual grounding architectures, we predict the target object and a chain of anchors in a causal manner. This chain of anchors is based on the logical sequence of steps a human will follow to reach the target. For instance, in Figure 1, to reach the `chair` target, we first have to localize the `white and red boxes`, then the `bookshelf`. By imitating the human learning process, we can devise a transparent and interpretable 3D framework that details the model's steps until localizing the target. To show that our proposed framework can be easily integrated into any existing architecture, we incorporated it into four different baselines: LAR (Bakr et al., 2022), SAT (Yang et al., 2021), MVT (Huang et al., 2022), and ViL (Chen et al., 2022). CoT3DRef achieves state-of-the-art results on Sr3D, Nr3D (Achlioptas et al., 2020), and ScanRefer (Chen et al., 2020) without requiring additional manual annotations by devising an efficient pseudo-label generator to provide inexpensive guidance to improve learning efficiency. Whereas it boosts the performance by 3.6%, 4%, 5%, 0.5% on Nr3D and 10%, 11%, 9%, 1% on Sr3D, respectively. Proper design of such an approach is pivotal in attaining a significant performance gain while circumventing the need for intensive human annotations. A pertinent example can be drawn from the labeling procedure employed in PhraseRefer (Yuan et al., 2022), which demanded a cumulative workforce commitment of 3664 hours, roughly equivalent to an extensive five-month timespan. Moreover, using additional manual annotations on the Nr3D dataset led to a noteworthy enhancement in referring accuracy, boosting the performance by 9% compared to the baselines: LAR, SAT, and MVT, respectively. Consequently, using ScanRefer, our approach surpasses SAT and MVT by 6.5% and 6.8%, respectively. In addition, as depicted in Figure 2, CoT3DRef shows a remarkable capability of learning from limited data, where training on only 10% of the data is enough to beat all the baselines, which are trained on the entire data. Our contributions are summarized as follows:

- We propose a 3D data-efficient Chain-of-Thoughts based framework, CoT3DRef, that generates an interpretable chain of predictions till localizing the target.
- We devise an efficient pseudo-label generator to provide inexpensive guidance to improve learning efficiency.
- Our proposed framework achieves state-of-the-art performance on Nr3D, Sr3D, and Scan-Refer benchmarks without requiring manually annotated data.
- Using 10% of the data, our framework surpasses the existing state-of-the-art methods.

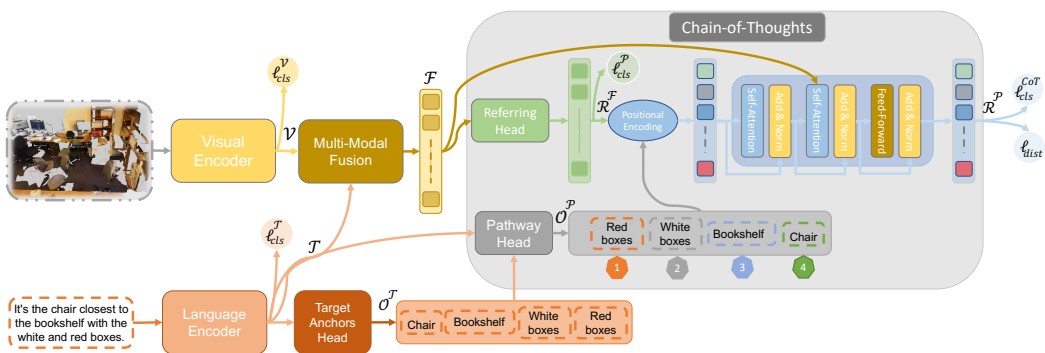

Figure 3: An overview of our Chain-of-Thoughts Data-Efficient 3D visual grounding framework (CoT3DRef). First, we predict the anchors $\mathcal{O}^{\mathcal{T}}$ from the input utterance, then sort the anchors in a logical order $\mathcal{O}^{\mathcal{P}}$ using the Pathway module. Then, we feed the multi-modal features $\mathcal{F}$, the parallel localized objects $\mathcal{R}^{\mathcal{F}}$, and the logical path $\mathcal{O}^{\mathcal{P}}$ to our Chain-of-Thoughts decoder to localize the referred object and the anchors in a logical order $\mathcal{R}^{\mathcal{P}}$.

## 2 RELATED WORK

**3D visual grounding.** Significant progress has been made in 3D visual grounding thanks to advancements in deep learning and computer vision, as well as the availability of grounded datasets (Achlioptas et al., 2020; Chen et al., 2020; Abdelreheem et al., 2022a; Yuan et al., 2022). One approach involves using graph-based models (Achlioptas et al., 2020; Feng et al., 2021; Yuan et al., 2021; Huang et al., 2021) to represent the scene as nodes and edges, while attention mechanisms help to focus on relevant parts of the scene. Another approach (Roh et al., 2022) is to convert the visual input into language tokens using a classification head. These tokens can then be combined with the input utterance and fed into a transformer architecture to learn the relationships between input sequence elements. Moreover, recent work (Bakr et al., 2022; Yang et al., 2021) explores distilling knowledge from 2D to 3D in a multi-view setup. However, none of the existing works models the explicit reasoning process behind the prediction of the target object.

**Chain of thoughts.** The chain-of-thought concept has been used in many different machine learning applications, including natural language processing (Wei et al., 2022; Chowdhery et al., 2022; Lyu et al., 2023; Wang et al., 2022; Zhang et al., 2023; Madaan & Yazdanbakhsh, 2022), and robotics (Jia et al., 2023; Yang et al., 2022). In the context of 3D Visual grounding, developing a chain-of-thought approach provides a natural way to explicitly model the grounding reasoning process, which to the best of our knowledge has not been explored. An extended version is discussed in the Appendix A.9.

## 3 COT3DREF

In this section, we propose a simple yet effective approach to decompose the referring task into multiple interpretable steps by modeling the problem as Seq2Seq, as shown in Figure 3. First, we briefly cover the general 3D visual grounding models' skeleton and its main components (Sec. 3.1). Then, we present our method in detail (Sec. 3.2). Finally, we detail our loss function (Sec. 3.4).

### 3.1 PRELIMINARIES

We built our framework in a generic way where it could be integrated into any state-of-the-art 3D visual grounding model. An arbitrary 3D visual grounding model mainly consists of three essential blocks; a Visual Encoder, a Language Encoder, and a Multi-Modal Fusion module.

**Visual Encoder.** An arbitrary 3D scene $S \in \mathbb{R}^{N \times 6}$ is represented by $N$ points with spatial and color information, i.e., XYZ, and RGB, respectively. Using one of the off-the-shelf 3D object detectors or manual annotations, we have access to the object proposals $\mathcal{P} = \{P_k\}_{k=1}^{L}$, where $P_k \in \mathbb{R}^{N' \times 6}$, $N'$ represents the number of the object's points and $L$ is the number of proposals in the scene. Then, the visual encoder encodes the proposals into lower-resolution feature maps $\mathcal{V} = \{V_k\}_{k=1}^{L}$, where $V_k \in \mathbb{R}^{1 \times d}$ and $d$ is the number of hidden dimensions.

**Language Encoder.** Simultaneously, given an input utterance that describes a particular object in a scene, called target, a pre-trained BERT model (Devlin et al., 2018) encodes the input sentence into $\mathcal{T} = \{T_j\}_{j=1}^{W}$, where $W$ is the maximum sentence length. Then, a language classification head

is utilized to predict only the referred object. We notice that limiting the language encoder to only predict the target object restricts its capability of learning representative features.

**Multi-Modal Fusion.** After encoding the object proposals and the utterance, a multi-modal fusion block is exploited to refine the visual features $\mathcal{V}$ based on the language embeddings $\mathcal{T}$ generating fused features $\mathcal{F} = \{F_k\}_{k=1}^{L}$, where $F_k \in \mathbb{R}^{1 \times d}$. Both graphs (Achlioptas et al., 2020) and transformers (Huang et al., 2022; Bakr et al., 2022; Yang et al., 2021) are explored to capture the correlation between the two different modalities. However, our CoT framework can be integrated easily into any existing 3D visual grounding architecture, regardless of how the multi-modal features $\mathcal{F}$ are obtained.

## 3.2 Chain-of-Thoughts

We decompose the referring task into multiple interpretable steps, whereas to reach the final target the model must first predict the anchors one by one, in a logical order called Chain-of-Thoughts. To this end, we first have to predict the anchors from the input utterance, then sort the anchors in a logical order using our Pathway module. Then, we replace the naive referring decoder with our Chain-of-Thoughts decoder.

We formulate the referring task as a Seq2Seq problem by localizing the anchors as an intermediate step. Instead of anticipating the target directly, we first predict the chain of anchors sequentially, then utilize them to predict the target.

**Pathway generation.** First, we extend the language head, i.e., Target Anchors Head, to extract both the target and the anchors $\mathcal{O}^{\mathcal{T}} = \{O_i^T\}_{i=1}^{M}$, where $M$ is the maximum number of objects in the sentence, depicted in the lower red part of Figure 3. We add a "no_obj" class to pad the output to the maximum length $M$. However, the predicted anchors are unsorted or sorted based on the occurrence order in the sentence, which can not be fit in our Chain-of-Thoughts framework. Accordingly, we introduce a "Pathway Head" which takes the encoded sentence $\mathcal{T}$ and the predicted objects of the utterance $\mathcal{O}^{\mathcal{T}}$ to produce logically ordered objects $\mathcal{O}^{\mathcal{P}} = \left(O_i^P\right)_{i=1}^{M}$. One possible solution is to exploit an MLP head to predict the logical order for each object. However, for better performance, we use a single transformer encoder layer to capture the correlation between different objects.

**Sequence-to-Sequence.** Similar to the language stream, we first employ a parallel referring head to localize the referred object alongside the anchors, ignoring their logical order. The parallel referring head only takes the multi-modal features $\mathcal{F}$ as input and localizes the target and the anchors $\mathcal{R}^{\mathcal{F}} = \{R_i^F\}_{i=1}^{M}$. We experiment both with localizing the target and the anchors in parallel, and localizing them one by one in a sequential logical order using previously localized objects as prior information. In other words, we formulate the 3D referring task as a Seq2Seq task, where the input is two sequences; 1) a set of object proposals $\mathcal{P}$, 2) a sequence of words (utterance). The output is a sequence of locations for the anchors and the target. Specifically, we add positioning awareness to the plain localized objects $\mathcal{R}^{\mathcal{F}}$ using the predicted ordered objects $\mathcal{O}^{\mathcal{P}}$, which act as a positional encoding. These positions indicate a logical order that mimics human perception (McVay & Kane, 2009; Chen et al., 2017). Furthermore, we employ a single transformer decoder layer, depicted in Figure 3, to localize the objects in sequence w.r.t the predicted logical order $\mathcal{O}^{\mathcal{P}}$. In this decoder layer, the queries are $\mathcal{R}^{\mathcal{F}} + \mathcal{O}^{\mathcal{P}}$ and the values and keys are $\mathcal{F}$. Accordingly, the attention maps denoted as $\mathcal{A}$ follow Eq. 1.

$$\mathcal{A} = \sigma\left(\frac{(\mathcal{R}^{\mathcal{F}} + \mathcal{O}^{\mathcal{P}})\mathcal{F}^T}{\sqrt{d}}\right), \tag{1}$$

where $\sigma$ is the Softmax function and $d$ is the embedding dimensions. We use a masked self-attention layer to enforce the Chain-of-Thoughts, where while predicting the next object's location, we only attend to the previously located objects. This could be interpreted as a one-direction CoT. We also experiment with another variant where no masking is applied so that we can attend to any object in the chain, as shown in Appendix A.6.

## 3.3 Pseudo labels

During the training phase, our proposed framework requires more information than the standard available GT in existing datasets (Achlioptas et al., 2020; Chen et al., 2020). These datasets only annotate the referred object and the target. However, our framework requires anchor annotations. Three types of extra annotations are needed: 1) Given an input utterance, we need to identify the

mentioned objects other than the target; the *anchors*. 2) Once we extract the anchors from the utterance, we need to know their logical order to create a chain of thoughts. 3) Finally, we need the localization information for each anchor, i.e., to assign a bounding box to every anchor.

To make our framework self-contained and scalable, we do not require any manual effort. Instead, we collect pseudo-labels automatically without any human intervention.

**Anchors parser.** We extract the textual information from the utterance using rule-based heuristics and a scene graph parser (Schuster et al., 2015; Wu et al., 2022). First, we extract the whole mentioned objects and their relations from the utterance using the scene graph parser. Then, we match the objects to their closest matching class from the ScanNet labels (Dai et al., 2017a) using SBERT (Reimers & Gurevych, 2019). Due to the free-from nature of Nr3D, the anchors mentioned in the GT descriptions sometimes do not precisely match the ScanNet class labels. For instance, the GT description is "The plant at the far right-hand side of the bookcase tucked in the furthest corner of the desk." However, there is no "bookcase" class in ScanNet. Therefore, we need to match it to the nearest ScanNet class labels, which in this case will be "bookshelf."

**Anchors pathway.** We utilize GPT-3.5 (Ouyang et al., 2022) to extract the logical order of objects given an input utterance, using in-context learning (Brown et al., 2020). The full prompt used for pathway extraction is provided in Appendix A.12.

**Anchors localization.** The anchors localization module employs object proposals $\mathcal{P}$, extracted relations $\mathcal{R}$ and utterance objects $\widetilde{\mathcal{O}^{\mathcal{T}}}$ to establish associations between anchors and object bounding boxes within an input scene. Our method involves iterating over all anchors extracted from the utterance and searching for candidate objects in $\mathcal{P}$ that share the same class. When the anchor class is represented singularly in the scene, we return the matched object. However, in scenarios where disambiguation is required due to multiple objects belonging to the anchor class, we leverage the parsed spatial relations and the localized objects within the scene to identify the intended anchor accurately, termed FIND in Algorithm 1. However, it is not guaranteed that the FIND function will be able to localize the remaining unlocalized anchors accurately. Thus, in this case, as shown in the last step in Algorithm 1, we randomly sample an object of the same class. We summarize our localization method in Algorithm 1.

---

**Algorithm 1** Localizing objects mentioned in an input utterance

---

**Input**:
- $\widetilde{\mathcal{O}^{\mathcal{T}}} = \{o_i\}_{i=1}^{K}$: unlocalized objects mentioned in the utterance, *extracted using the syntactic parser*.
- $\mathcal{P}$: object proposals for the scene $\mathcal{P} = (\mathcal{B}, \mathcal{C})$, with:
  $\mathcal{B} = \{\mathbf{b_i}\}_{i=1}^{L}$: bounding boxes $\mathbf{b}_i \in \mathbb{R}^6$ and $\mathcal{C} = \{c_i\}_{i=1}^{L}$ classes of the $L$ object proposals for the scene.
- RELATE : $\widetilde{\mathcal{O}^{\mathcal{T}}} \to \mathcal{R} \times \widetilde{\mathcal{O}^{\mathcal{T}}}$: map objects mentioned in the input utterance to a ⟨spatial relation, object⟩ pair.
- FIND : $\mathcal{P} \times \mathcal{R} \times \mathcal{P}^m \to \mathcal{P}$: map a localized object, a relation and a set of $m$ candidate objects to an output localized object.

**Output**:
- $\mathcal{A}$: localized anchors used in the utterance, where $\mathcal{A} \subseteq \mathcal{P}$.

1: **function** LOCALIZE($o, \mathcal{P}, \mathcal{A}$)                                                              ▷ Localizing a single utterance
2:    $\mathcal{K} \leftarrow \{\mathcal{P}_j \mid o \propto c_j, c_j \in \mathcal{C}\}$                                ▷ Find a set of candidates with the same class as $o$
3:    **if** $|\mathcal{K}| = 1$ **then**                                                                            ▷ A single object has the anchor class
4:        **return** $\mathcal{K}$
5:    **for** $(r_m, o_k) \in$ RELATE($o$) **do**
6:        **if** $o_k \in \mathcal{A}$ **then**: **return** FIND($\mathcal{P}_k, r_m, \mathcal{K}$)                   ▷ The related object is already localized
7:        $p \leftarrow$ LOCALIZE($o_k, \mathcal{P}, \mathcal{A}$)                                                    ▷ Otherwise, attempt to localize the related object
8:        **if** $p \neq \emptyset$ **then**:    **return** $\{p\} \cup$ FIND($p, r_m, \mathcal{K}$)
9:    **return** $\{p\}, p \sim \{p_i \mid p_i = (\mathbf{b_i}, c_i), o \propto c_i, p_i \in \mathcal{P}\}$            ▷ If all else fails: randomly sample an object of the same class
10: $\mathcal{A} \leftarrow \emptyset$
11: **for** $o_i \in \widetilde{\mathcal{O}^{\mathcal{T}}}$ **do**: $\mathcal{A} \leftarrow \mathcal{A} \cup$ LOCALIZE($o_i, \mathcal{P}, \mathcal{A}$)

---

## 3.4 LOSSES

Mainly, three losses exist in most existing 3D visual grounding architectures. Two of them are considered auxiliary losses, i.e., 3D classification loss $\mathcal{L}_{cls}^{\mathcal{V}}$ and language classification loss $\mathcal{L}_{cls}^{\mathcal{T}}$. Hence, the third one is the primary loss, i.e., referring loss $\mathcal{L}_{ref}$. First, we extend the language classification loss $\mathcal{L}_{cls}^{\mathcal{T}}$ by recognizing the referred object class and the anchors based on the input utterance. Similarly, the referring loss is extended to localize both the target and the anchors, termed as parallel referring loss $\mathcal{L}_{ref}^{\mathcal{P}}$, as we localize the target and anchors in one step. Furthermore, we add another referring loss after the transformer decoder, termed as CoT referring loss $\mathcal{L}_{ref}^{\mathcal{COT}}$. Finally, an

Table 1: Ablation study for different components of our CoT3DRef framework. First, we compare the baseline, i.e., MVT (Huang et al., 2022), against the parallel and the chain-of-thoughts approaches. Then, we show the effect of adding distractor loss. All the experiments are conducted using the Nr3D dataset (Achlioptas et al., 2020).

| | Data Percentage | +Distractor. Loss. | +Parallel | +CoT | Nr3D ↑ | Sr3D ↑ |
|---|---|---|---|---|---|---|
| (a) | 10% | - | - | - | 27.6 | 48.8 |
| (b) | 10% | - | ✓ | - | 31.7 | 55.3 |
| (c) | 10% | - | - | ✓ | 37.5 | 65.2 |
| (d) | 10% | ✓ | - | ✓ | **38.2** | **66.4** |
| (e) | 100% | - | - | - | 55.2 | 66.0 |
| (f) | 100% | - | ✓ | - | 57.0 | 71.5 |
| (g) | 100% | - | - | ✓ | 60.0 | 72.7 |
| (h) | 100% | ✓ | - | ✓ | **60.4** | **73.2** |

Table 2: Ablation study for different numbers of the transformer blocks used in our CoT decoder, based on MVT.

| # Transformer Blocks | Nr3D ↑ | Sr3D ↑ |
|---|---|---|
| 1 | **60.4** | **73.2** |
| 2 | 60.4 | 73.3 |
| 4 | 60.1 | 72.9 |

Table 3: Ablation study to highlight the effect of the anchors' quality on the final target referring accuracy, based on Nr3D dataset and MVT as a baseline.

| Method | Anchors Ref. Acc. | Target Ref. Acc. |
|---|---|---|
| Baseline | N/A | 55.1 |
| +CoT +Zeroing Anchor Loss | 4.5 | 55.0 |
| +CoT +Pseudo Labels | 72.3 | 60.4 |
| +CoT +Human Labels | **73.6** | **64.4** |

auxiliary distractor binary classification loss $\mathcal{L}_{dist}$ is introduced to distinguish between the target and the distractors, i.e., objects with the same class as the target. Grouping all these losses, we optimize the whole model in an end-to-end manner with the following loss function:

$$\mathcal{L} = (\lambda_{\mathcal{V}} \cdot \mathcal{L}_{cls}^{\mathcal{V}}) + (\lambda_{\mathcal{T}} \cdot \mathcal{L}_{cls}^{T}) + \lambda_{ref} \cdot (\mathcal{L}_{ref}^{\mathcal{P}} + \mathcal{L}_{ref}^{\mathcal{COT}}) + \lambda_{dist} \cdot \mathcal{L}_{dist}, \tag{2}$$

where $\lambda$ is the corresponding loss weight for each term.

## 4 EXPERIMENTAL RESULTS

**Datasets.** To probe the effectiveness of our proposed framework, CoT3DRef, we conduct evaluations on three 3D visual-grounding benchmarks, namely Nr3D, Sr3D (Achlioptas et al., 2020) and ScanRefer (Chen et al., 2020). Nr3D contains 41.5K natural, free-form utterances gathered from humans through a referring game, while Sr3D consists of 83.5K synthetic utterances. Consequently, ScanRefer provides 51.5K utterances of 11K objects for 800 3D indoor scenes.

**Network Configuration.** We model the Pathway module using only one transformer encoder layer, and the CoT decoder using a single transformer decoder layer. The number of heads used are 7 and 16 for the Pathway module and CoT decoder, respectively. The number of proposals $L$ and the maximum sentence length $W$ are 52 and 24, respectively. $L$ and $W$ define the sizes of the input sequences to our CoT decoder. The maximum number $M$ of objects in the sentence, the output sequence length for our CoT decoder, is 8 and 3 for Nr3D and Sr3D, respectively. Following previous works (Abdelreheem et al., 2022b; Achlioptas et al., 2020; He et al., 2021; Roh et al., 2022; Jain et al., 2021; Yang et al., 2021; Qi et al., 2017), we randomly sample 1024 points for each proposal, set the hidden dimensions $d$ to 768, and train the model for 100 epochs from scratch using the weight initialization strategy described in (He et al., 2015). The initial learning rate is set to $10^{-4}$ and decreases by 0.65 every ten epochs. The Adam optimizer (Kingma & Ba, 2014) and a mini-batch size of 24 per GPU are used for training all the models. We set the losses weights as follows: $\lambda_{\mathcal{V}} = 5$, $\lambda_{\mathcal{T}} = 0.5$, $\mathcal{L}_{ref} = 5$, and $\lambda_{dist} = 1$. We used the PyTorch framework and a single NVIDIA A6000 GPU for training.

### 4.1 ABLATION STUDIES

We conducted several ablation studies to validate each module in our framework, termed CoT3DRef.

**CoT vs. Parallel.** To assess our CoT3DRef framework, we have to disentangle the CoT from the pseudo label generation module. In other words, it could be thought that the achieved gain caused by just accessing additional supervision signal, i.e., anchors' annotations. To this end, we implement a parallel approach, that has access to the anchors' labels, however, localizes the targets and anchors in one shot without any interaction between them. In contrast, our CoT3DRef framework leverages the causality between the anchors and the target through our chain-of-thoughts decoder. As shown in Table 1, on the challenging setup, where we assume access for only 10% of the training data while testing on the entire testing dataset, the parallel variant boosts the performance by 4% and 6.5% over the vanilla MVT using Nr3D and Sr3D, respectively (row b). On the other hand, our CoT3DRef framework surpasses the vanilla MVT by 10% and 16.4% using Nr3D and Sr3D, respectively (row c). Consequently, using the entire data, our CoT surpasses both the parallel and the vanilla MVT approaches by 3% and 5% on Nr3D, and by 1% and 6.7% on Sr3D, respectively.

Table 4: Benchmarking results on Nr3D and Sr3D datasets (Achlioptas et al., 2020). We emphasize that we did not require any additional GT annotations, in contrast to SAT (Yang et al., 2021) which requires access to real 2D images, and PhraseRefer (Yuan et al., 2022) and ScanEnts (Abdelreheem et al., 2022a) require manual annotations for the whole anchors in the data. We report the standard-deviation $\sigma$ in green.

| Method | GT Anchors | Nr3D ↑ | | | | | Sr3D ↑ | | | | |
|---|---|---|---|---|---|---|---|---|---|---|---|
| | | Overall($\sigma$) | Easy | Hard | View-dep. | View-indep. | Overall($\sigma$) | Easy | Hard | View-dep. | View-indep. |
| ReferIt3D (Achlioptas et al., 2020) | N/A | 35.6 | 43.6 | 27.9 | 32.5 | 37.1 | 40.8 | 44.7 | 31.5 | 39.2 | 40.8 |
| Text-Guided-GNNs (Huang et al., 2021) | N/A | 37.3 | 44.2 | 30.6 | 35.8 | 38.0 | 45.0 | 48.5 | 36.9 | 45.8 | 45.0 |
| InstanceRefer (Yuan et al., 2021) | N/A | 38.8 | 46.0 | 31.8 | 34.5 | 41.9 | 48.0 | 51.1 | 40.5 | 45.4 | 48.1 |
| 3DRefTrans (Abdelreheem et al., 2022b) | N/A | 39.0 | 46.4 | 32.0 | 34.7 | 41.2 | 47.0 | 50.7 | 38.3 | 44.3 | 47.1 |
| 3DVG-Trans (Zhao et al., 2021) | N/A | 40.8 | 48.5 | 34.8 | 34.8 | 43.7 | 51.4 | 54.2 | 44.9 | 44.6 | 51.7 |
| FFL-3DOG (Feng et al., 2021) | N/A | 41.7 | 48.2 | 35.0 | 37.1 | 44.7 | - | - | - | - | - |
| TransRefer3D (He et al., 2021) | N/A | 42.1 | 48.5 | 36.0 | 36.5 | 44.9 | 57.4 | 60.5 | 50.2 | 49.9 | 57.7 |
| LanguageRefer (Roh et al., 2022) | N/A | 43.9 | 51.0 | 36.6 | 41.7 | 45.0 | 56.0 | 58.9 | 49.3 | 49.2 | 56.3 |
| 3D-SPS (Luo et al., 2022) | N/A | 51.5 | 58.1 | 45.1 | 48.0 | 53.2 | 62.6 | 56.2 | 65.4 | 49.2 | 63.2 |
| LAR (Bakr et al., 2022) | N/A | 48.9 | 56.1 | 41.8 | 46.7 | 50.2 | 59.4 | 63.0 | 51.2 | 50.0 | 59.1 |
| +CoT3DRef (**Ours**) | ✗ | **52.5**±0.1 | **59.4** | **45.7** | **50.4** | **53.6** | - | - | - | - | - |
| +CoT3DRef (**Ours**) | ✓ | **58.0**±0.2 | **65.1** | **50.9** | **54.6** | **60.0** | **69.0**±1.0 | **74.2** | **59.4** | **58.9** | **70.1** |
| SAT (Yang et al., 2021) | N/A | 49.2 | 56.3 | 42.4 | 46.9 | 50.4 | 57.9 | 61.2 | 50.0 | 49.2 | 58.3 |
| +PhraseRefer (Yuan et al., 2022) | ✓ | 54.4 | 62.1 | 47.0 | 51.2 | 56.0 | 63.0 | 66.2 | 55.4 | 48.7 | 63.6 |
| +ScanEnts (Abdelreheem et al., 2022a) | ✓ | 52.5 | 59.8 | 45.6 | 51.3 | 53.2 | - | - | - | - | - |
| +CoT3DRef (**Ours**) | ✗ | **53.1**±0.2 | **60.3** | **46.2** | **51.8** | **53.9** | - | - | - | - | - |
| +CoT3DRef (**Ours**) | ✓ | **58.1**±0.4 | **65.1** | **51.6** | **57.0** | **58.4** | **69.1**±0.5 | **71.9** | **62.8** | **54.5** | **70.2** |
| MVT (Huang et al., 2022) | N/A | 55.1 | 61.3 | 49.1 | 54.3 | 55.4 | 64.5 | 66.9 | 58.8 | 58.4 | 64.7 |
| +ScanEnts (Abdelreheem et al., 2022a) | ✓ | 59.3 | 65.4 | 53.5 | 57.3 | 60.4 | - | - | - | - | - |
| +PhraseRefer (Yuan et al., 2022) | ✓ | 59.0 | - | - | - | - | 68.0 | - | - | - | - |
| +CoT3DRef (**Ours**) | ✗ | **60.4**±0.2 | **66.2** | **54.6** | **58.1** | **61.5** | - | - | - | - | - |
| +CoT3DRef (**Ours**) | ✓ | **64.4**±0.2 | **70.0** | **59.2** | **61.9** | **65.7** | **73.2**±0.5 | **75.2** | **67.9** | **67.6** | **73.5** |
| ViL3DRel (Chen et al., 2022) | N/A | 63.6 | 70.0 | 56.9 | 61.3 | 64.1 | 72.8 | 72.5 | 66.3 | 61.5 | 72.8 |
| +CoT3DRef (**Ours**) | ✗ | **64.1**±0.1 | **70.4** | **57.5** | **61.7** | **64.8** | - | - | - | - | - |
| +CoT3DRef (**Ours**) | ✓ | **64.0**±0.2 | **70.4** | **57.3** | **61.5** | **64.8** | **73.6**±0.5 | **73.0** | **67.2** | **61.9** | **73.7** |

Table 5: Benchmarking results while training jointly on Nr3D and Sr3D datasets (Achlioptas et al., 2020).

| Method | Nr3D+Sr3D ↑ | | | | |
|---|---|---|---|---|---|
| | Overall($\sigma$) | Easy | Hard | View-dep. | View-indep. |
| ReferIt3D (Achlioptas et al., 2020) | 37.2 ± 0.3 | 44.0 ± 0.6 | 30.6 ± 0.3 | 33.3 ± 0.6 | 39.1 ± 0.2 |
| non-SAT (Yang et al., 2021) | 43.9 ± 0.3 | - | - | - | - |
| TransRefer3D (He et al., 2021) | 47.2 ± 0.3 | 55.4 ± 0.5 | 39.3 ± 0.5 | 40.3 ± 0.4 | 50.6 ± 0.2 |
| SAT (Yang et al., 2021) | 53.9 ± 0.2 | 61.5 ± 0.1 | 46.7 ± 0.3 | 52.7 ± 0.7 | 54.5 ± 0.3 |
| MVT (Huang et al., 2022) | 58.5 ± 0.2 | 65.6 ± 0.2 | 51.6 ± 0.3 | 56.6 ± 0.3 | 59.4 ± 0.2 |
| MVT+CoT3DRef (**Ours**) | **62.5**±0.4 | **70.0**±0.5 | **53.0**±0.2 | **58.3**±0.1 | **63.9**±0.3 |

Table 6: Benchmarking results on Scan-Refer dataset (Chen et al., 2020).

| Method | Data Percentage | | | |
|---|---|---|---|---|
| | 10% | 40% | 70% | 100% |
| MVT (Huang et al., 2022) | 36.4 | 51.5 | 54.5 | 57.4 |
| +PhraseRefer (Yuan et al., 2022) | - | - | - | 59.9 |
| +CoT3DRef (**Ours**) | **48.6** | **60.1** | **62.5** | **64.2** |
| SAT (He et al., 2021) | 34.9 | 49.3 | 50.5 | 53.8 |
| +PhraseRefer (Yuan et al., 2022) | - | - | - | 57.5 |
| +CoT3DRef (**Ours**) | **49.1** | **57.7** | **58.9** | **60.3** |

**Distractor Loss.** We add an auxiliary distractor binary classification loss $\mathcal{L}_{dist}$ to disambiguate the target and the distractors, as discussed in Sec. 3.4. As shown in Table 1, rows d and h, incorporating it boosts the performance by 0.5-1%.

**Anchors Quality Effect.** To establish the affirmative role of anchors in refining target localization accuracy without inducing detrimental effects, we conducted a worst-case simulation. In this simulation, all anchors were falsely detected. In other words, we deliberately zero the localization loss associated with anchors during training, causing the decoder to randomly and inaccurately predict anchor locations. Remarkably, the target accuracy remained unaltered, as shown in Table 3, reflecting the robustness of the approach. The accuracy experienced a decrement of approximately 5%, dropping from 60.4% to 55%. Encouragingly, even in this scenario, the accuracy mirrored the baseline performance, steadfast at 55.1%. This substantiates that while anchor detection may exhibit inaccuracies, the broader framework's efficacy in target localization remains largely unaffected. In contrast, we replaced the pseudo labels with manual annotations (Abdelreheem et al., 2022a). This substitution serves as an upper-bound reference point for evaluation. As shown in Table 3, the exchange of noisy pseudo labels with precise manual annotations led to a noteworthy 4% enhancement in referring accuracy for Nr3D, elevating it from 60.4% to 64.4%.

**Number of Transformer Blocks.** As shown in Table 2, we have explored using 1, 2, and 4 transformer blocks in our CoT referring decoder. However, we didn't notice a significant gain in performance; therefore, we preferred to use a single transformer block.

## 4.2 COMPARISON TO STATE-OF-THE-ART

We verify the effectiveness of our proposed framework, termed CoT3DRef on three well-known 3D visual grounding benchmarks, i.e., Nr3D, Sr3D (Achlioptas et al., 2020) and ScanRefer (Chen et al., 2020). By effectively localizing a chain of anchors before the final target, we achieve state-of-the-art results without requiring any additional manual annotations. As shown in Table 4, when we integrate our module into four baselines; LAR (Bakr et al., 2022), SAT (Yang et al., 2021), MVT (Huang et al.,

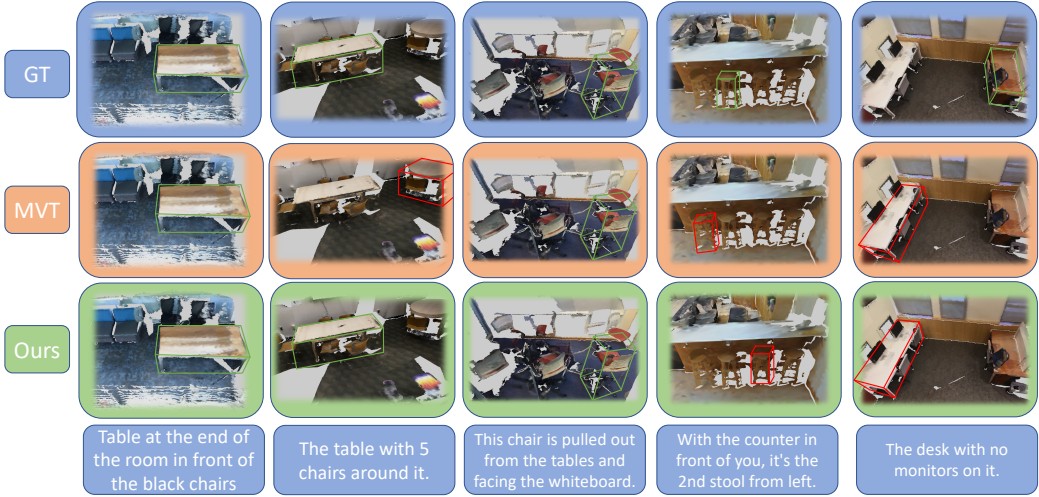

Figure 4: Our qualitative results. First row indicates the GT w.r.t the input utterance, demonstrated in the last row. Second and third rows show the qualitative results for MVT and our method, respectively. The success and the failure cases are shown in green and red boxes, respectively.

2022) and ViL (Chen et al., 2022), it boosts the accuracy by 3.6%, 4%, 5%, 0.5% on Nr3D and 10%, 11%, 9%, 1% on Sr3D, respectively. The disparity between the gain achieved on Nr3D and Sr3D is due to our pseudo label module that hinders achieving more gain on Nr3D. Exchanging of our noisy pseudo labels with precise manual annotations led to a noteworthy enhancement in referring accuracy, where our module boosts the performance by 9% compared to the baselines; LAR, SAT and MVT, respectively. This outcome underscores our model's ability to yield enhanced performance not only for simpler descriptions (Sr3D) but also in the context of more intricate, free-form descriptions (Nr3D). We have a limited gain on only ViL. A detailed analysis that justifies this behaviour is mentioned in the Appendix A.4.

**Nr3D+Sr3D.** In addition, we jointly train on Nr3D and Sr3D. Whereas, we augment the Nr3D training data with Sr3D, while testing on the same original Nr3D test-set. Consistently with the previous results mentioned in Table 1 and 4, we surpass all the existing work by 3.5%. As shown in Table 5, we achieve 62.5% grounding accuracy, while MVT only achieves 58.5% accuracy.

**ScanRefer.** To further show the effectiveness of our proposed method, we have conducted several experiments on the ScanRefer dataset across different data percentages on MVT and SAT baselines. As shown in Table 6, we outperform both MVT and SAT by a significant margin across the entire data percentages, i.e., 10%, 40%, 70%, and 100%. More specifically, integrating our CoT framework into MVT has boosted the performance by 12.2%, 8.6%, 8%, and 6.8%, respectively. In addition to MVT, we have integrated our CoT framework into the SAT baseline, where the same performance gain has been achieved, probing our method's effectiveness across a comprehensive range of baseline models, datasets, and different available data percentages.

**Data Efficiency.** To further validate the effectiveness of our framework, we assess it on a more challenging setup, where we assume access to only limited data. Four percentage of data is tested, i.e., 10%, 40%, 70%, and 100%. As shown in Figure 2, on the Sr3D dataset, using only 10% of the data, we match the same performance of MVT and SAT that are trained on 100% of the data. This result highlights the data efficiency of our method. Furthermore, when trained on 10% of the data on Nr3D with noisy pseudo labels (Sec. 3.3), we still surpass all the baselines with considerable margins.

**Qualitative results.** As shown in Figure 4, the first three examples show that our model successfully localizes the referred objects by leveraging the mentioned anchors, such as "the table with 5 chairs around". However, in the ambiguous description shown in the fourth example: "2nd stool from the left", the model incorrectly predicts the stool, as it is view-dependent. In other words, if you look at the stools from the other side, our predicted box will be correct. Additionally, the last example shows a challenging scenario where a negation in the description is not properly captured by our model.

# 5 Discussion and Limitations

**Comparison with PhraseRefer (Yuan et al., 2022) and ScanEnts (Abdelreheem et al., 2022a).** We acknowledge the great work proposed by PhraseRefer (Yuan et al., 2022) and ScanEnts (Abdelreheem et al., 2022a), which paved the way for the importance of paying attention not just to the target object but also to the anchors. Thus, underlying approaches have some similarities in terms of demonstrating the importance of including anchors in the pipeline. However, there are a lot of significant differences: 1) Design a CoT framework for 3D visual grounding that explicitly models causal reasoning while interpreting the instruction inspired by the human perception system. 2) Show that we can design an efficient framework that achieves state-of-the-art results on three challenging benchmarks, i.e., Nr3D, Sr3D (Achlioptas et al., 2020), and ScanRefer (Chen et al., 2020), without requiring human labels. Proper design of such an approach is pivotal in attaining a good performance while circumventing the need for intensive human annotations. A pertinent example can be drawn from the labeling procedure employed in PhraseRefer (Yuan et al., 2022), which demanded a cumulative workforce commitment of 3664 hours, roughly equivalent to an extensive five-month timespan.

**Pseudo labels accuracy.** The accuracy of the pseudo-labels plays a vital role in the overall performance. To evaluate its performance, we manually collect ground-truth labels for 1) the predicted orderings of the anchors in the utterance and 2) the final localization of the anchors as predicted by the geometry module, based on 10% of the Nr3D. To evaluate orderings predicted by in-context learning, we use the normalized Levenshtein edit distance between two sequences, where a length of 1 means that every object in the sequence is incorrect. We achieve an average distance of $0.18$ between predicted and ground-truth orderings. We consider anchor-wise localization accuracy to evaluate the geometry module's accuracy, where the anchors considered for each sequence are those mentioned in the input utterance. We achieve a 77 % accuracy for this task compared to human annotators. Overall, a significant accuracy gap is measured between automatically collected pseudo-labels and ground-truth data, contributing to the performance loss observed on the Nr3D dataset.

**Pseudo module limitations.** Despite designing a data efficient framework that could be integrated into any existing 3D visual grounding architecture, and achieve SOTA results without requiring any manual annotation efforts, our pseudo label module hinders achieving more gain on Nr3D. Accordingly, we encourage the future efforts to try to enhance the pseudo module performance. In addition, the anchor localization block in our pseudo module is tailored on ScanNet dataset (Dai et al., 2017a), and will thus need some adaptations to be usable on other 3D scene datasets.

**Pathway module limitations.** Our Pathway module is responsible of generating a chain of logical order for the extracted objects from the input utterance. However, it does not handle the multi-path scenario, where multiple paths are valid. For instance, given this utterance "It is the chair besides the desk, which has a book and a lamp above it", we have two possible starting points, i.e., locating the lamp first or the book. Thus, for simplicity, we start by the last mentioned object in the utterance, in the multiple paths scenario. Nevertheless, one possible solution to handle this limitation implicitly through building a graph that reasons the different possibilities (Salzmann et al., 2020).

# 6 Conclusion

We propose CoT3DRef: a novel and interpretable framework for 3D visual grounding. By formulating the problem of 3D visual grounding from a natural language instruction as a sequence-to-sequence task, our approach predicts a chain of anchor objects that are subsequently utilized to localize the final target object. This sequential approach enhances interpretability and improves overall performance and data efficiency. Our framework is data-efficient and outperforms existing methods on the Nr3D and Sr3D datasets when trained on a limited amount of data. Furthermore, our proposed chain-of-thoughts module can easily be integrated into other architectures. Through extensive experiments, we demonstrate consistent performance gains over previous state-of-the-art methods operating on Referit3D. Importantly, our approach does not rely on any additional manual annotations. Instead, we leverage automatic rule-based methods, syntactic parsing, and in-context learning to collect pseudo-labels for the anchor objects, thereby eliminating the laborious and time-consuming process of manually annotating anchors. Overall, our work advances 3D visual grounding by making a step towards bridging the gap between machine perception and human-like understanding of 3D scenes.

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

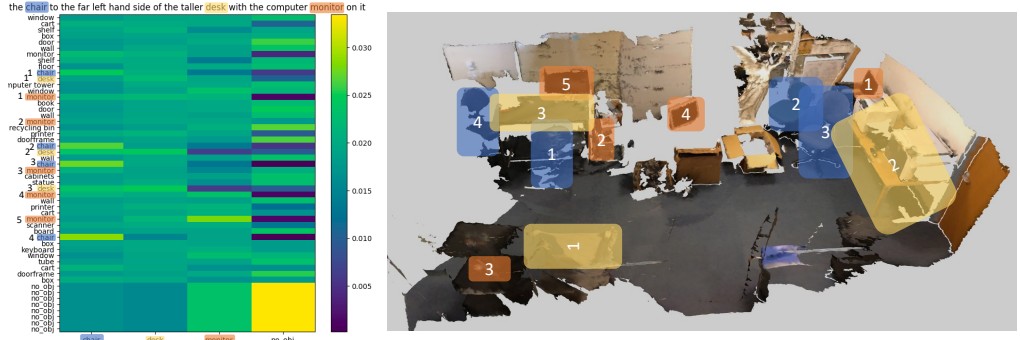

Figure 5: Identification of failure cases as a benefit of the interpretability. In this example, there are two anchors mentioned in the description, desk and monitor, and the target is the chair. The correct chair should be number two, however, the model predicts number four. By visualizing the attention maps, on the left, we can identify the main cause of the wrong prediction, whereas, the first anchor localize a wrong desk (desk #3). Therefore, the rest of the chain, i.e., the monitor and the chair are localized wrongly.

# A    APPENDIX

Our Appendix contains the following sections:

- Identification of failure cases as a benefit of the interpretability.
- Broader Impact and Ethical Considerations.
- GPT-generated path influence on the overall grounding accuracy.
- Justification for CoT performance when integration into Vil3DRef.
- Do we use additional supervision?
- Causal vs. Non-Causal CoT.
- Nr3D Ambiguity.
- Related Work.
- Explore data-augmentation effect in the limited data scenario (10%).
- Pseudo labels human-evaluation.
- In-context prompt used for object order extraction.
- Examples of generated paraphrases for an utterance.

## A.1    IDENTIFICATION OF FAILURE CASES AS A BENEFIT OF THE INTERPRETABILITY

To demonstrate our interpretability abilities, we visualize attention maps as shown in Figure 5. In Figure 5, the input description is "the chair to the far left hand side of the taller desk with the computer monitor on it." Accordingly, there are two anchors mentioned in the description, i.e., desk and monitor, and the target is the chair. The correct chair should be number two, however, the model predicts number four. By visualizing the attention maps, on the left part of Figure 5, we can identify the main cause of the wrong prediction, whereas, the first anchor localize a wrong desk (desk #3) instead of desk #2. Therefore, the rest of the chain, i.e., the monitor and the chair are localized wrongly. This example also shows the ambiguity in Nr3D, where it is hard to say which desk is the taller, desk #2 or desk #3.

## A.2    BROADER IMPACT AND ETHICAL CONSIDERATIONS

3D visual grounding holds profound implications across diverse domains and applications, spanning from integrating outdoor systems, e.g., autonomous driving and indoor navigation systems.

**Outdoor navigation systems:** 3D visual grounding empowers the agents to execute a broad spectrum of tasks, facilitating assistive systems catering to the needs of elderly individuals and those

with disabilities—such as the visually impaired. For instance, in the context of visually impaired individuals, the technology's ability to ground objects within their 3D environment from language descriptions is pivotal in navigation tasks such as determining the nearest exit or chair.

**Autonomous driving:** In the realm of autonomous driving, a 3D visual grounding system takes on a critical role in translating navigational instructions into actionable responses within the dynamic visual landscape. This technology enables seamless communication between the language input, such as navigation commands, and the 3D world the autonomous vehicle operates in. For instance, when given a directive like "merge into the right lane at the upcoming intersection where the three pedestrians are walking," the 3D visual grounding system interprets this command by analyzing the spatial layout, recognizing lanes and pedestrians, and identifying the appropriate merging point. By bridging the gap between language instructions and the complex visual cues inherent in driving scenarios, this system enhances the vehicle's ability to accurately interpret and execute instructions, contributing to safer and more efficient autonomous driving experiences.

**Data-efficiency importance for autonomous driving:** Introducing a data-efficient solution for the 3D visual grounding task holds immense importance in the context of autonomous driving, particularly considering the challenges associated with manually labeling vast amounts of 3D point cloud data from lidar sensors. The conventional approach of manually annotating such data is labor-intensive and economically burdensome. By proposing a data-efficient solution, our system addresses a critical bottleneck in developing autonomous driving technology. This allows the autonomous vehicle to learn and generalize from minimal labeled data, significantly reducing the dependency on large, expensive datasets. The capacity to make accurate 3D visual groundings with limited labeled information not only streamlines the training process but also makes deploying autonomous vehicles more scalable and cost-effective.

**Federated learning for indoor navigation systems:** Aligned with this imperative, the inherent data-efficient nature of our approach positions it as an ideal candidate for federated learning schemes, which is perfectly aligned with the indoor setup when privacy is required, e.g., the robot is operating in a personalized environment such as the client home. This becomes particularly relevant when a robot is introduced into a novel indoor environment and necessitates rapid learning from minimal interactions with its new user without sending the private data of the new environment to the server; thus, learning from limited data is essential.

## A.3   GPT-generated path influence on the overall grounding accuracy

The GPT-generated path is only used as a teacher label for training a simple MLP that predicts the path during the inference time. Therefore, GPT is not involved in the system during inference, as we train a simple MLP layer, i.e., the Pathway-Head in Figure 3, to predict the logical order. For instance, when the system is deployed on a robot, it will be efficient as our simple MLP Pathway-Head is utilized to generate a reasonable order based on the input description.

Accordingly, GPT could be seen as a design choice to obviate the need for human annotations during training.

In Table 7, the Pseudo and GT labels are for the anchors' localization, not the path. For the path, we always use our MLP-predicted path. However, the accuracy will not be affected if we replace it with the GPT-generated path. As shown in Table 7, we achieve 64.4% and 64.5% using our predicted and GPT-generated paths, respectively. This is an intuitive result, as when we assess the performance of the Pathway Head separately, we get 98% accuracy w.r.t the GPT-generated path. Therefore, replacing the Pathway Head generated path by the GPT ones is expected not to impact the final performance.

## A.4   Justification for CoT performance when integration into ViL3DRef

For ViL, our observations, in Figure 2, reveal a substantial performance enhancement in the demanding data-efficient scenario, accompanied by marginal improvements when employing the entire dataset. These modest gains are likely due to nearing the peak attainable accuracy of the dataset, influenced by the limitations posed by annotation errors.

To substantiate our hypothesis, we have conducted three experiments:

Table 7: Ablation study about the GPT-generated path influence on the overall grounding accuracy.

| Method | Grounding Accuracy |
|---|---|
| MVT | 55.1 |
| MVT+ScanEnts | 59.3 |
| MVT+PhraseRefer | 59.0 |
| MVT+CoT(Pseudo labels) | 60.4 |
| MVT+CoT(GT labels) | 64.4 |
| MVT+CoT(GT labels+GPT Path) | 64.5 |

Table 8: Comparison between causal and non-causal masking in our CoT transformer decoder-based, based on MVT (Huang et al., 2022).

| +Pre-trained P++ | +Teacher Loss | ViL | Ours |
|---|---|---|---|
| ✗ | ✗ | 59.4 | **70.3** |
| ✓ | ✗ | 69.5 | **72.7** |
| ✓ | ✓ | 72.8 | **73.6** |

- Replacing our pseudo labels with GT: We replaced our pseudo labels with ground truth annotations (Yuan et al., 2022; Abdelreheem et al., 2022a), which is intuitively guaranteed to enhance performance. As shown in Table 4, the substitution of our pseudo labels with ground truth annotations yielded performance improvements across the three alternative baselines; LAR (Bakr et al., 2022), SAT (Yang et al., 2021), MVT (Huang et al., 2022), with the exception of ViL.
- Deactivation of ViL's Knowledge Distillation Loss (Teacher Loss): We disabled the knowledge distillation loss proposed by ViL. This maneuver, as shown in Table 8, while significantly impairing ViL's performance, exhibited negligible influence on ours (ViL+CoT).
- Deactivation of P++ pre-training: Consequently, the deactivation of the teacher loss alongside P++ pre-training had a pronounced detrimental effect on ViL's performance, yielding a referring accuracy of 59.4%. In contrast, our own performance was only slightly affected, maintaining a referring accuracy of 70.3%. These findings collectively serve to reinforce our hypothesis that our performance within the ViL architecture is effectively limited by an upper bound.

### A.5  DO WE USE ADDITIONAL SUPERVISION?

Our main thesis is to show that the design of our CoT framework can be accomplished with notable efficiency, obviating the need for human intervention. Furthermore, we emphasize its demonstrable capacity for substantial enhancement of existing methodologies, particularly within the context of data efficiency, as shown in Figure 2. To this end, we have devised an efficient pseudo-label generator to provide inexpensive guidance to improve the learning efficiency of most existing methods, as we demonstrated in our experiments in Table 4 and Figure 2. These pseudo labels are not required at test time. Proper design of such an approach is pivotal in attaining a significant performance gain while circumventing the need for intensive human annotations. A pertinent example can be drawn from the labeling procedure employed in PhraseRefer (Yuan et al., 2022), which demanded a cumulative workforce commitment of 3664 hours, roughly equivalent to an extensive five-month timespan. Hence, our pseudo-label generation module is delineated into three core components: the anchor parser, the anchor localization module, and the pathway prediction module. Notably, it is essential to highlight that solely the pathway prediction component leverages a pre-trained model to generate a ground truth but is not involved during training or at inference. More specifically, the anchor parser and localization modules do not contain any learnable parameters nor pre-trained models, as they use rule-based heuristics where only two pieces of information are needed: 1- objects' class labels. 2 - objects' position. These two sources of supervision are utilized by all referring architectures, where object class labels are used as ground truth for the classification head, and object positions are incorporated in the model's pipeline either by concatenation with the objects' point clouds or input to the multi-modal transformer as a positional encoding. Accordingly, both

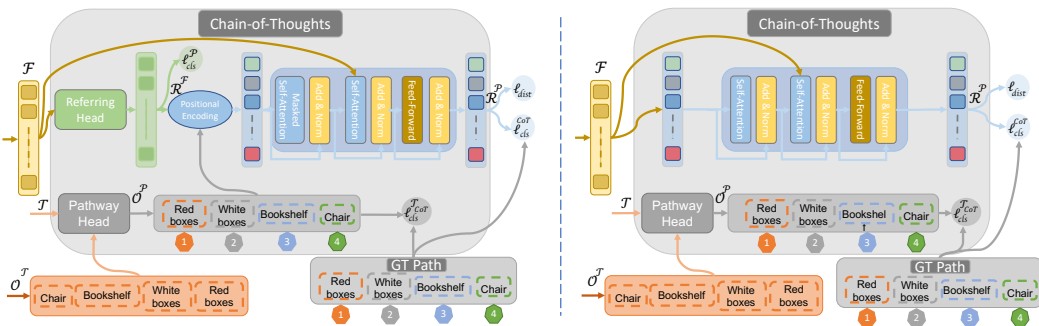

Figure 6: Comparison between two variants of our Chain-of-Thoughts framework. On the left is the causal version, where we first localize the anchors in parallel (shown in green), then feed them to the masked transformer decoder to predict the refined anchors' location in auto-regressive manner. In contrast, on the right is the non-causal variant of our CoT framework, where we feed the multi-modal features $F$ directly to the transformer decoder and remove the causal constrains, i.e., all the anchors can attend to each others. In both variants, the decoder outputs the anchors in auto-regressive way and respects the GT logical path for the chain of anchors.

Table 9: Comparison between causal and non-causal masking in our CoT transformer decoder-based, based on MVT (Huang et al., 2022).

| Masking Type | Nr3D ↑ | Sr3D ↑ |
|---|---|---|
| Causal | 60.4 | 73.2 |
| Non-Causal | **60.9** | 73.2 |

modules do not require any additional supervision. Solely, the anchor pathway prediction module requires a pre-trained model to generate the ground truth path, but it is not involved during training or at inference. The trajectory of reasoning (referred to as the Chain-of-Thoughts) produced by the pathway prediction module is indispensable within our framework. The omission of the anchor chain would inevitably result in the degradation of our framework, resembling the parallel scenario wherein anchors and the target object are independently detected. Conversely, our framework adopts a sequential, auto-regressive approach, wherein each anchor is predicted consecutively. To show the effectiveness of our proposed method, we compare it against the parallel approach, as shown in Table 1. In summary, solely the pathway prediction component leverages a pre-trained model to generate a ground truth but is not involved during training or at inference. Additionally, with zero human efforts, we achieve a significant performance gain on four distinct baselines, as shown in Figure 2.

### A.6 CAUSAL VS. NON-CAUSAL COT

In contrast to the conventional teacher-forcing technique, our approach employs a parallel referring head to predict the initial, unordered locations of anchors, depicted in green within Figure 3. This implementation facilitates an investigation into the implications of causal masking, by removing the causality constraints. Essentially, this entails permitting the decoder to encompass a broader scope by attending to all anchors concurrently, irrespective of their order. However, our CoT decoder still adhere to the correct sequential order for prediction of both anchors and the target object auto-regressively. Figure 6, illustrates the modifications made to the architecture to accommodate the random masking approach. Empirical results, as shown in Table 9, indicate a marginal enhancement in accuracy on the Nr3D dataset—almost 0.5%—upon relaxation of the causality constraints. However, no discernible improvements materialize on the Sr3D dataset.

### A.7 NR3D AMBIGUITY

The challenges posed by Nr3D are not solely attributed to its free-form nature; rather, it is compounded by its susceptibility to ambiguity. A visual representation of this issue is provided in Figure 7 showcasing instances of ambiguity where even human annotators encounter difficulties in accurately

With the counter in front of you, it's the 2nd stool from left.

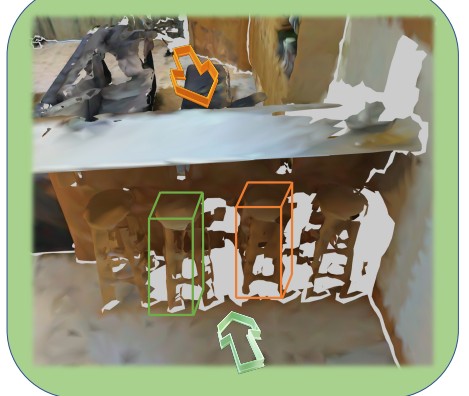

Two chair sitting together and one in right side is green.

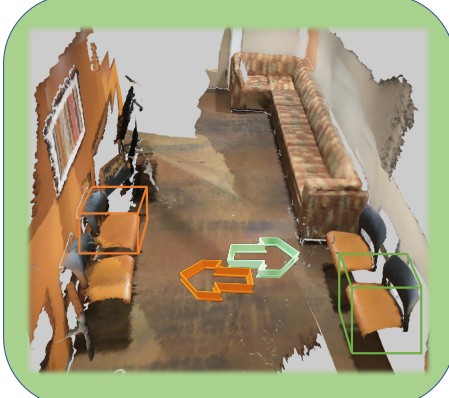

Looking at the table with two monitors, pick the one on the right.

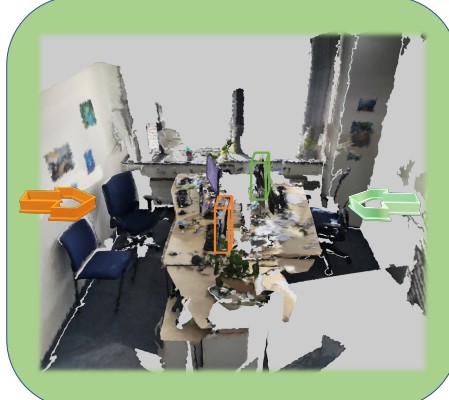

The leftmost pillow in the set.

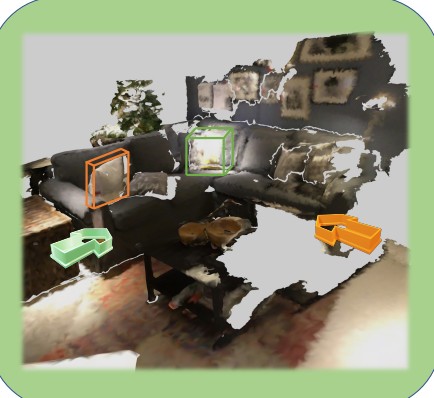

Figure 7: A visual representation of Nr3D ambiguity view-dependent issue, where predictions hinge on the speaker's viewpoint. For instance, these four samples show that given an input description two options could be considered true based on the speaker's viewpoint. The green and orange arrows represent two different views. The color of the box indicates its corresponding view-point.

localizing objects due to viewpoint-dependent considerations. In such cases, correct predictions hinge on the speaker's viewpoint, an aspect that often remains undisclosed.

To fortify this standpoint, we conducted an in-depth analysis of the consistency between two recently released manually annotated datasets (Yuan et al., 2022; Abdelreheem et al., 2022a). Surprisingly, the matching accuracy between these annotations stands at a mere 80%, signifying the extent of inherent data ambiguity and the ensuing challenges. In essence, divergent annotations from distinct human annotators due to ambiguity impedes the potential for further enhancements.

## A.8 ANCHOR LOCALIZATION ACCURACY

For each scene, we have the objects' proposals $\mathcal{P} = (\mathcal{B}, \mathcal{C})$, where $\mathcal{B} = \{\mathbf{b_i}\}_{i=1}^{L}$ bounding boxes $\mathbf{b}_i \in \mathbb{R}^6$, $\mathcal{C} = c_{i_{i=1}}^{L}$ is the class label and L is the number of objects in the scene. Given the extracted objects' names, $\widetilde{\mathcal{O}^{\mathcal{T}}} = \{o_i\}_{i=1}^{K}$, using our syntactic parser, the anchors localization module automatically assign each object's name to the corresponding box, $\widetilde{\mathcal{A}} = (\widetilde{\mathcal{B}}, \widetilde{\mathcal{O}^{\mathcal{T}}})$, where $\widetilde{\mathcal{B}} \subseteq \mathcal{B}$ and $\widetilde{\mathcal{A}} \subseteq \mathcal{P}$. To assess the assignation quality, we ask the annotators to assign a box for each extracted object's name $\widetilde{\mathcal{O}^{\mathcal{T}}}$ from the input utterance, creating the GT set $\mathcal{A} = (\bar{\mathcal{B}}, \widetilde{\mathcal{O}^{\mathcal{T}}})$, where $\bar{\mathcal{B}} \subseteq \mathcal{B}$ and $\mathcal{A} \subseteq \mathcal{P}$. Then, we measure the precision between the two sets, i.e., $\widetilde{\mathcal{A}}$ and $\mathcal{A}$.

## A.9 RELATED WORK

Our novel architecture draws success from several areas, including 3D visual grounding, chain-of-thought reasoning, and scene graph parsers.

**3D visual grounding.** Existing work can be grouped into one- and two-stage approaches. Single-stage methods depend on detecting objects directly by fusing text features with point-cloud level visual representations and enable grounding multiple objects from a single sentence (Kamath et al., 2021; Luo et al., 2022). Two-stage methods dissociate the object detection task from selecting a target among detected objects (Achlioptas et al., 2020; Chen et al., 2020), by detecting objects during the first stage and then classifying the resulting objects. To benchmark the 3D visual grounding task, ReferIt3D (Achlioptas et al., 2020) and ScanRefer (Chen et al., 2020) were introduced by collecting textual annotations from the ScanNet dataset (Dai et al., 2017a). Early work depended on graph-based approaches (Veličković et al., 2018; Wang et al., 2019) to infer spatial relations. The object graph is constructed by connecting each object with its top nearest neighbors (Abdelreheem et al., 2022b; Feng et al., 2021; Huang et al., 2021; Yuan et al., 2021) based on Euclidean distances. Taking advantage of the Transformer's attention mechanism, which is naturally suitable for multi-modal features fusion, (He et al., 2021; Zhao et al., 2021) achieved remarkable performance by applying self-attention and cross-attention on input features. While methods such as BEAUTY-DETR (Jain et al., 2022) and (Luo et al., 2022) adopt a single-stage approach, most works follow the two-stage approach with pre-detected object proposals. LAR (Bakr et al., 2022), and SAT (Yang et al., 2021) use a multi-modal approach by introducing 2D images of the scenes to aid object grounding. Conversely, MVT (Huang et al., 2022) focuses on 3D point clouds by extracting multiple views of the objects and aggregating them to improve view robustness to make the transformer view-invariant. LanguageRefer (Roh et al., 2022) converts the visual grounding task into a language-based object prediction problem instead of the cross-modal approach. Similarly, NS3D (Hsu et al., 2023) focuses on language features by translating language into hierarchical programs. EDA (Wu et al., 2022) explicitly separates textual attributes and performs dense alignment between language and point cloud objects. ViL3DRel (Chen et al., 2022) includes a spatial self-attention layer that considers the relative distances and orientations between objects in 3D point clouds.

**Chain of thoughts.** In machine learning, a chain of thought refers to the series of intermediate steps that a model takes to arrive at its final decision or prediction. The chain-of-thought concept has been used in many different machine learning applications, including natural language processing (Wei et al., 2022; Chowdhery et al., 2022; Lyu et al., 2023; Wang et al., 2022; Zhang et al., 2023; Madaan & Yazdanbakhsh, 2022), and robotics (Jia et al., 2023; Yang et al., 2022). In natural language processing, chain-of-thought prompting adds intermediate reasoning steps to the output of LLMs, which leads to increased reasoning capabilities (Zhao et al., 2023; Wei et al., 2022). In robotics, understanding the chain of thought can help to identify which sensory inputs were used to arrive at a particular action or decision. Moreover, (Jia et al., 2023) divides the primary goal into subgoals to tackle the complex

Table 10: Ablation study for different components of our CoT3DRef framework, such as different augmentation techniques and distractor loss. All the experiments are conducted on 10% of Nr3D dataset.

|  | +Scene Aug. | +Point Aug. | +Lang. Aug. | +Ref. Aug. | Ref. Acc. |
|---|---|---|---|---|---|
| (a) | - | - | - | - | 27.6 |
| (b) | - | - | ✓ | - | 37.5 |
| (c) | ✓ | - | - | - | 34.5 |
| (d) | - | ✓ | - | - | 34.0 |
| (e) | - | ✓ | ✓ | - | 35.5 |
| (f) | - | - | ✓ | ✓ | **37.5** |

sequential decision-making problems using the Hierarchical RL (HRL) (Hutsebaut-Buysse et al., 2022).

**Scene graph parser.** Extracting scene graphs is a fundamental task in NLP and multimodal vision. This problem depends on extracting directed graphs, where nodes are objects and edges represent relationships between objects (Lou et al., 2022; Xu et al., 2017; Wu et al., 2019; Jiang et al., 2016; Yang et al., 2018; Schuster et al., 2015). Scene graphs have been used in scene understanding to enhance image captioning and predicting pairwise relations between detected objects (Li et al., 2017b; Dai et al., 2017b; Li et al., 2017a). Other works have leveraged graphs to enhance text-to-image generation models by learning the relations between the objects in the sentence (Feng et al., 2023; Johnson et al., 2018). Existing off-the-shelf tools (Schuster et al., 2015) can parse the language description grammatically into a scene graph with a dependency tree (Wu et al., 2022).

### A.10 EXPLORE DATA-AUGMENTATION EFFECT IN THE LIMITED DATA SCENARIO (10%)

**Visual-based Augmentation.** Two visual augmentations are explored at the scene and point cloud levels. At the scene level, we randomly shuffle every object presented in the scene with another object of the same class from one of the ScanNet scenes. At the point cloud level, we apply a small random isotropic Gaussian noise, rotate the points along each axis, and randomly flip the $x$ and $y$ axes. In addition, we apply a small random RGB translation over color features. These visual augmentations are used to enhance the robustness and generalization capabilities of our architecture.

**Language-based Augmentation.** We utilize a T5 (Raffel et al., 2019) transformer model fine-tuned on paraphrase datasets to augment the input utterances. To ensure the quality of the generated paraphrases, we use the Parrot library (Damodaran, 2021) to filter them based on a fluency and adequacy score. This aims to enhance the naturalness and diversity of the generated utterances. We showcase an example of an input utterance alongside its corresponding generated paraphrases in Table 11.

**Referring-based Augmentation.** The maximum number of objects in the sentence $M$ is 8 and 3 for Nr3D and Sr3D, respectively. However, in 52% and 98.5% of Nr3D and Sr3D, respectively, two objects only are mentioned in the utterance; the target and one anchor. This motivates us to apply a simple referring-based augmentation by swapping the target and the anchor while keeping the same semantic information. For instance, if the relation between two objects is "near to", it should be the same after swapping. However, if it was "on the right of", it should be swapped to be "on the left of".

**Results.** We have explored several augmentation techniques to enhance our architecture's robustness and generalization capabilities. However, we report the results on only 10% of the data as we notice no gain is achieved when leverage more training data, e.g., 40%. As shown in Table 10, the language and referring augmentations (rows b and f) increase the performance slightly. However, the two visual-based augmentations (rows c, d, and e) do not give any performance gain.

### A.11 PSEUDO LABELS HUMAN-EVALUATION

The accuracy of the pseudo-labels plays a vital role in the overall performance. To evaluate pseudo-labels, we manually collect ground-truth labels for:

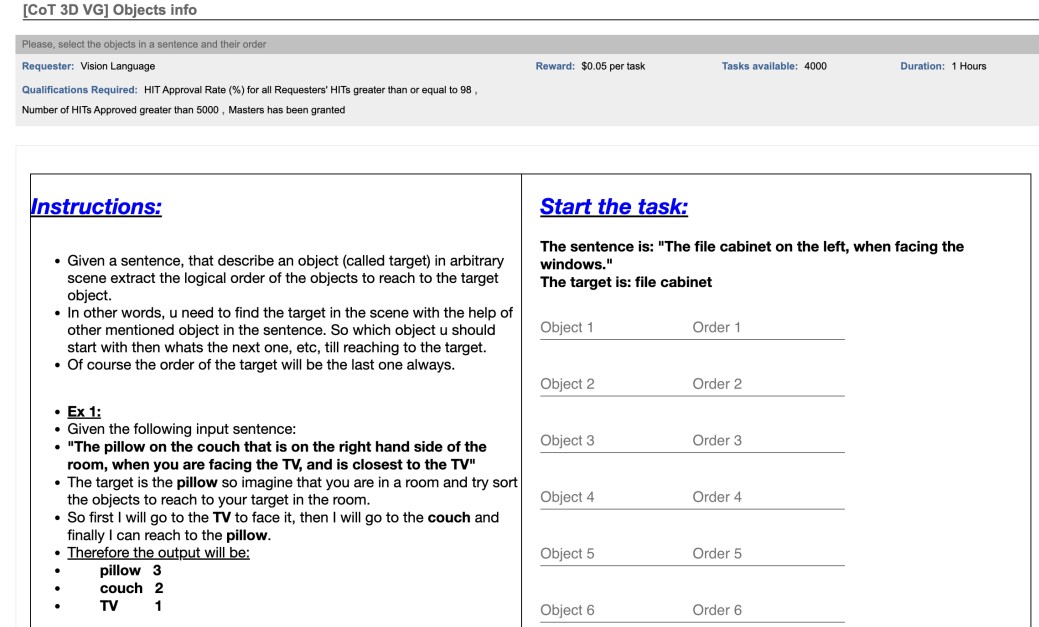

Figure 8: Our User-Interface (UI), designed on Amazon-Mechanical-Turk (AMT) (Crowston, 2012). We detailed the instructions on the left, and the task is mentioned on the right. The annotators are asked to extract the mentioned objects in the input sentence in chronological order, then assign each object a logical order to reach the end goal, the target. We demonstrated the instructions on the left for each sample for better performance.

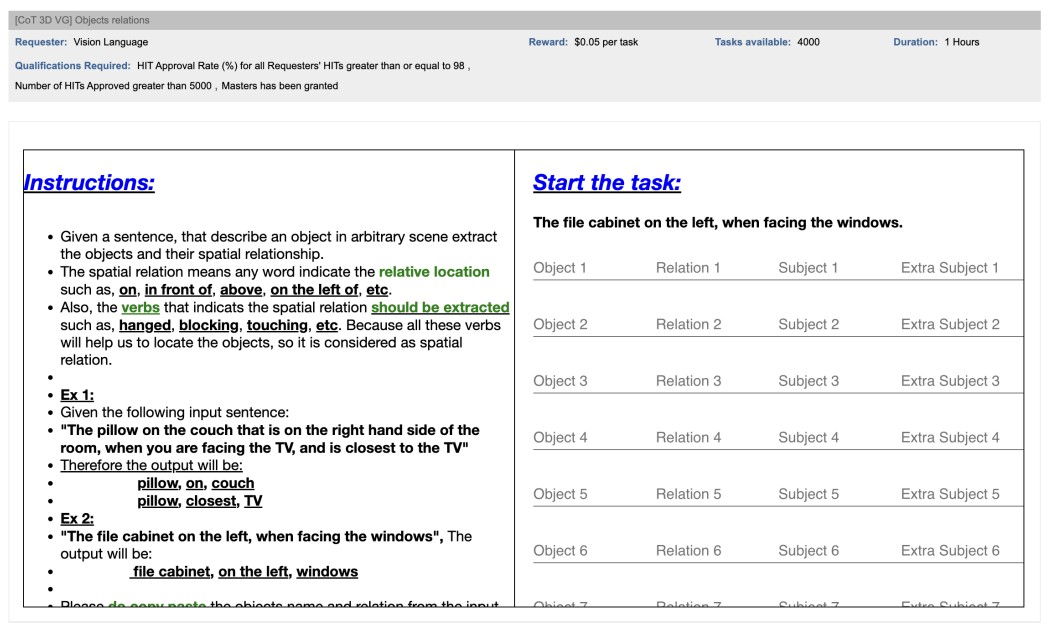

Figure 9: Our User-Interface (UI), designed on Amazon-Mechanical-Turk (AMT) (Crowston, 2012). We detailed the instructions on the left, and the task is mentioned on the right. The annotators are asked to chronologically extract the mentioned objects in the input sentence and their subject and relation in triplet format. We demonstrated the instructions on the left for each sample for better performance.

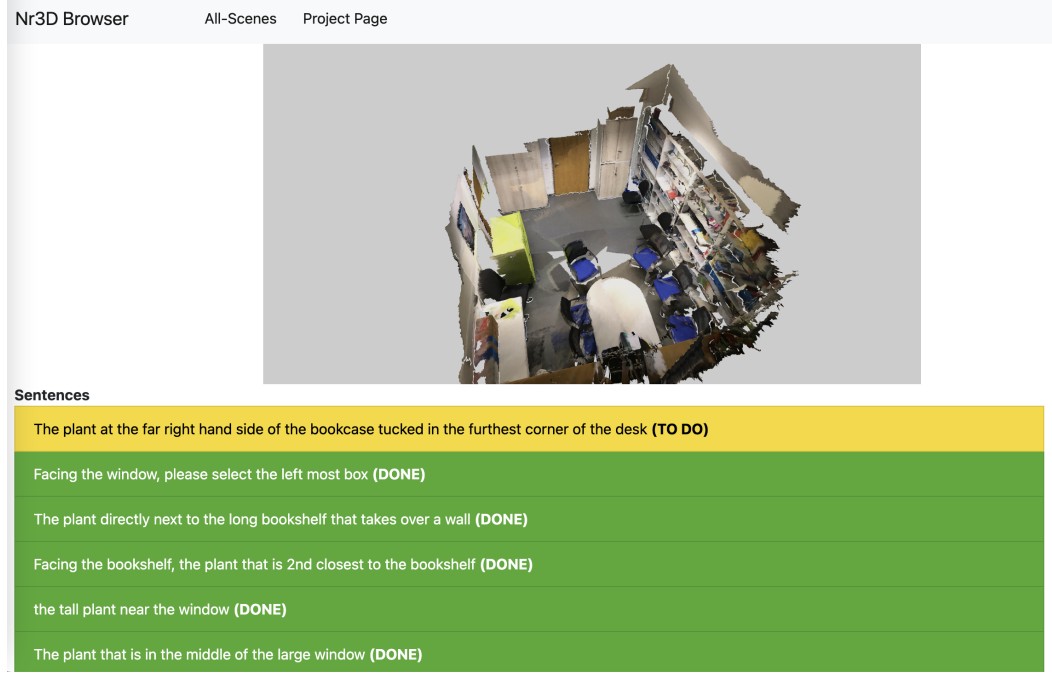

| Nr3D Browser | Project Page |
|---|---|
| scene0525_00 (19/75) | |
| scene0265_00 (1/41) | |
| scene0586_00 (0/38) | |
| scene0668_00 (3/49) | |
| scene0505_00 (0/226) | |
| scene0262_00 (0/51) | |
| scene0164_00 (0/56) | |
| scene0074_00 (0/13) | |
| scene0138_00 (0/38) | |
| scene0565_00 (0/115) | |
| scene0147_00 (0/64) | |
| scene0296_00 (0/150) | |
| scene0462_00 (1/16) | |

Figure 10: The intro screen of our web-based User-Interface (UI) for anchors localization task. On the intro page, we list the available scenes and the corresponding number of sentences for each scene in parentheses. More specifically, for each scene, we demonstrate the number of annotated sentences divided by the total number of sentences in that scene.

Figure 11: Our web-based User-Interface (UI) for anchors localization tasks. After clicking on a particular scene from the intro page, the 3D interactive window will be displayed at the top, and corresponding sentences will be displayed below it. To facilitate the annotators' task, we mark the annotated sentences in green and the remaining in yellow.

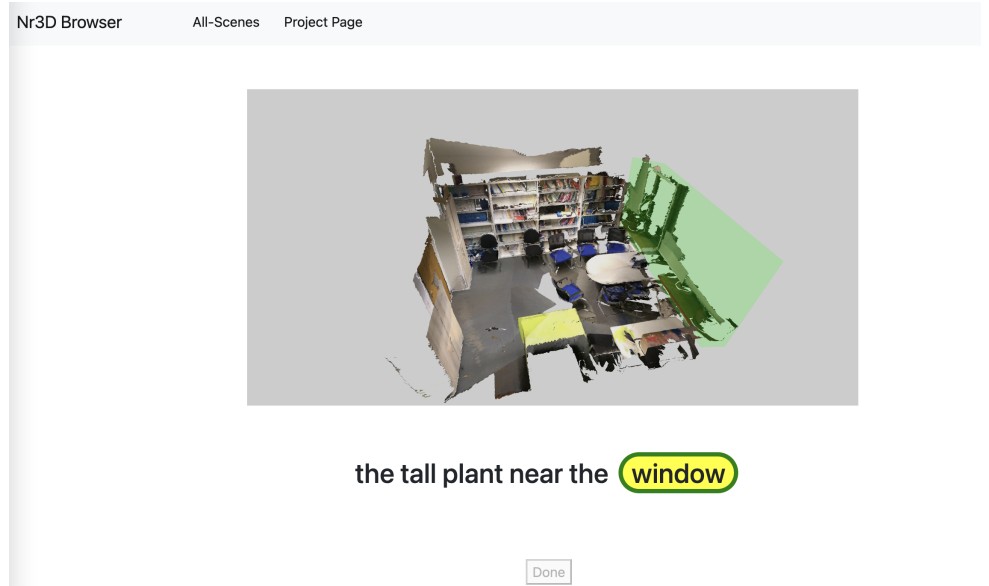

Figure 12: Our web-based User-Interface (UI) for anchors localization tasks. After clicking on a specific sentence, the 3D interactive window will be displayed at the top, and the corresponding sentence will be displayed below it. To facilitate the annotators' task, we mark the anchors' words in a clickable yellow box. Once a box is clicked, we show the corresponding boxes with the same class label as the clicked word. Finally, the annotator's task is to pick the correct box out of the colored boxes. In this example, we have a unique box for the word window; therefore, only one box is shown.

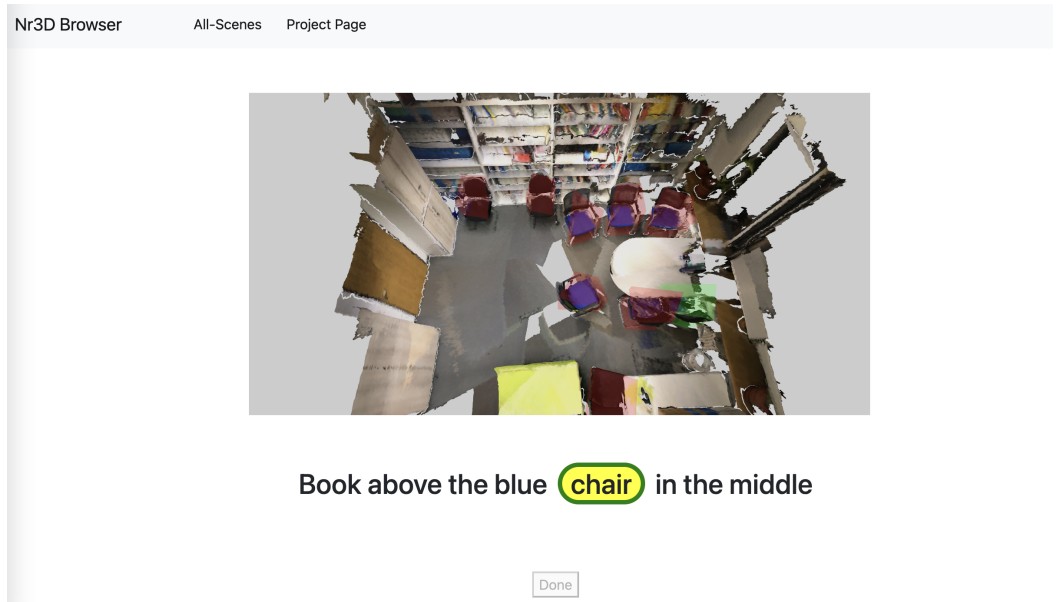

Figure 13: Our web-based User-Interface (UI) for anchors localization tasks. After clicking on a particular sentence, the 3D interactive window will be displayed at the top, and the corresponding sentence will be displayed below it. To facilitate the annotators' task, we mark the anchors' words in a clickable yellow box. Once a box is clicked, we show the corresponding boxes with the same class label as the clicked word. Finally, the annotator's task is to pick the correct box out of the colored boxes. In this example, we have nine boxes for the word chair. The selected box by the annotator is marked in green, while the other distractors are marked in red.

1. The predicted orderings of the anchors in the utterance. Figures 8 and 9 depict our User-Interface (UI), designed on Amazon-Mechanical-Turk (AMT) (Crowston, 2012) for objects' order and relations collecting. We detailed the instructions on the left, and the task is mentioned on the right. The annotators are asked to extract the mentioned objects in the input sentence in chronological order and their spatial relationship, then assign each object a logical order to reach the end goal, the target. We demonstrated the instructions on the left for each sample for better performance. To evaluate orderings predicted by in-context-learning, we use the normalized Levenshtein edit distance between two sequences, where a distance of 1 means that every object in the sequence is incorrect. We achieve an average distance of 0.18 between predicted and ground-truth orderings.

2. The final localization of the anchors as predicted by the geometry module, as demonstrated in Figures 10, 11, 12, and 13. To evaluate the geometry module's accuracy, we consider anchor-wise localization accuracy. Therefore, we design a web-based interface to collect ground-truth data for 10% of Nr3D dataset. At the entry screen, Depicted in Figure 10, we list the available scenes and the corresponding number of sentences for each scene in parentheses. More specifically, for each scene, we demonstrate the number of annotated sentences divided by the total number of sentences in that scene. To facilitate the annotators' task, we mark the annotated sentences in green and the remaining in yellow. Afterwards, when the annotator clicks on a particular scene, the 3D interactive window will be displayed at the top, and corresponding sentences will be displayed below it, as shown in Figure 11. We mark the anchors' words in a clickable yellow box. Once a box is clicked, we show the corresponding boxes with the same class label as the clicked word. Finally, the annotator's task is to pick the correct box out of the colored boxes. For instance in Figure 12, we have a unique box for the word window; therefore, only one box is shown. In contrast, Figure 13 shows more challenging sample, where we have nine boxes for the word chair. The selected box by the annotator is marked in green, while the other distractors are marked in red.

## A.12 IN-CONTEXT PROMPT USED FOR OBJECT ORDER EXTRACTION

We provide below the abridged prompt we use to extract the logical order of objects to follow to find the target object, following a given input utterance. We find empirically that 1) providing a large number of examples, 2) mentioning reasoning explanations, 3) asking to first provide the list of mentioned objects, and 4) repeating instructions at the end of the prompt increases the quality of the predicted object orderings produced by GPT3.5.

---

Given a set of instructions to locate a target object in a room, your task is to extract the order of objects mentioned in each instruction.
1. First list all the unordered mentioned objects in the sentence using the format: "**L:** [object_1, object_2, object_3]".
2. Then order the objects using the format "**R:** [object order]" and mark the target object with "t". Only include the names of the objects.
Additionally, please exclude any additional information or descriptors such as color or size adjectives. The target is the subject of the sentence.

**Q:** "The pillow on the couch that is on the right hand side of the room, when you are facing the TV, and is closest to the TV"
**L:** [TV, couch, pillow]
**R:** [1: TV, 2: couch, t: pillow]
(explanation: we first need to find the TV to orient ourselves,
then find the couch on the right hand-side of the room, and then the pillow on the couch. Do not mention the explanations in the future.)
**Q:** "The pillow on the bed that is on the far end of the room and is at the rear and right hand side of the bed"
**L:** [pillow, bed]
**R:** [1: bed, t: pillow]
**Q:** "Standing at the end of the bed looking towards the pillows, choose the pillow that is in the front, smaller and more to the right."
**L:** [bed, pillow]
**R:** [1: bed, t: pillow]
**Q:** "There are two groups of kitchen cabinets; three along the wall with the stove, and two on the opposite wall.
Looking at the wall opposite the stove, the one with the refrigerator, please select the kitchen cabinet closest to the refrigerator."
**L:** [kitchen cabinet, wall, stove, refrigerator]
**[...]**

Do not mention explanations. Only include objects in the list.
Do not add any text beyond the reply. The target is the subject of the sentence.
Mention all relevant objects in the reply. Don't use color or size adjectives.
Please exclude any color adjectives in your response. Avoid using any descriptive words related to color in your response.
**Query:**

---

## A.13 EXAMPLE OF GENERATED PARAPHRASES

In Table 11 below, we provide an example of an input utterance sampled from Nr3D and its associated generated paraphrases.

Table 11: Example of an input utterance from Nr3D and its associated paraphrases.

| | |
|---|---|
| **Input Utterance** | When entering the room, look on the nightstand to the left of the bed headboard, you will find some books. |
| **Paraphrase 1** | You'll find a few books on a nightstand just to the left of the headboard. |
| **Paraphrase 2** | You'll find a few books on a nightstand just to the left of the bed headboard. |
| **Paraphrase 3** | Look at the nightstand to the left of the headboard of the bed, and you will find some books. |
| **Paraphrase 4** | When you enter the room, look at the nightstand to the left of the headboard of the bed. |

