# OpenReview forum: "CoT3DRef: Chain-of-Thoughts Data-Efficient 3D Visual Grounding"
_ICLR.cc/2024/Conference — ICLR 2024 poster_

### Official Review · Reviewer_kjGR · 2023-10-15

**Soundness:** 3 good
**Presentation:** 3 good
**Contribution:** 3 good
**Rating:** 6
**Confidence:** 3

**Summary:**

The paper proposes an interpretable 3D visual grounding framework based on Chain-of-Thought in large language models, i.e., CoT3DReF.
It is a sequence-to-sequence (Seq2Seq) model by first predicting a chain of anchors and then the final target.
To provide supervision on the intermediate chaining, the paper devises a pseudo-label generator.
The proposed framework achieves state-of-the-art performance and shows its data efficiency on several 3D visual grounding benchmarks.

**Strengths:**

1. The paper integrate the idea of Chain-of-Thoughts into the 3D groudning tasks, and propose a 3D data-efficient and interpretable  framework CoT3DRef.
2. The technical contribution is sound, including Seq-2-Seq formulation and pseudo labels generation.
3. The proposed framework achieves state-of-the-art performance on 3D visual grounding tasks and shows its data efficiency.

**Weaknesses:**

In general, I am in favor of the paper, which makes an interesting first attempt toward CoT 3D grounding tasks. However, there are still some important while unexplored problems.

1. To make CoT3DRef self-contained and scalable, it is important to generate accurate pesudo labels. Although it has been discussed in Sec.5, I have some remaining questions/concerns:
- In Alg.1, FIND function maps a localized object, a relation and a set of candidate objects to an output localized object. What is the detailed process of FIND? Is the relations between objects known?
- Alg.1 receive the input of object proposals P. Is it the output from the 3D object detection model which requires extra training? If so, the anchor localization may be bottlenecked by the object proposals.
- Considering the elaborate process of psuedo label generation, the propose frameworks seems not "data efficient" in data preparation, though it is data efficient in 3D grounding training.

2. The interpretability is limited. From the input end, GPT-3.5 is used for extract the logical order of objects given an input utterance, thus adding interpretability. But the logical relations of the objects is not explicit modeled. Instead, it is implicit learned in the Seq-2-Seq network.

**Questions:**

See "Weaknesses".

---

> ### Author Response · Authors · 2023-11-21
>
> We thank you for your valuable and thoughtful feedback.
> We are encouraged that you find our paper novel, effective, and reasonable and that our experiments are comprehensive with state-of-the-art results.
> Below, we will address your concern and incorporate all the feedback.
>
>
> > 1: In Alg.1, FIND function maps a localized object, a relation and a set of candidate objects to an output localized object. What is the detailed process of FIND? Is the relations between objects known?
>
> The relations between objects are not known. Thus, we have to extract them.
> First, we extract the entire mentioned objects and their relations from the utterance using the scene graph parser [[1]](https://aclanthology.org/W15-2812.pdf), [[2]](https://openaccess.thecvf.com/content/CVPR2023/papers/Wu_EDA_Explicit_Text-Decoupling_and_Dense_Alignment_for_3D_Visual_Grounding_CVPR_2023_paper.pdf).
>
> Our method involves iterating over all anchors extracted from the utterance and searching for candidate objects in P that share the same class.
> When the anchor class is represented singularly in the scene, we return the matched object.
> However, we run the FIND function in scenarios where disambiguation is required due to multiple objects belonging to the anchor class.
> The FIND function leverages the parsed spatial relations and the localized objects, so far, within the scene to identify the remaining anchors accurately.
> However, it is not guaranteed that the FIND function will be able to localize the remaining unlocalized anchors accurately.
> Thus, in this case, as shown in the last step in Algorithm 1, we randomly sample an object of the same class.
>
> > 2: Alg.1 receive the input of object proposals P. Is it the output from the 3D object detection model? If so, the anchor localization may be bottlenecked by the object proposals.
>
> The prominent role of the localization module is to assign an object proposal for each anchor mentioned in the input description without requiring human effort.
> A pertinent example can be drawn from the labeling procedure employed in PhraseRefer, which demanded a cumulative workforce commitment of 3664 hours.
>
> Thus, we use the GT object proposals P from the ScanNet dataset, following the related works, e.g., [ReferIt3D](https://www.ecva.net/papers/eccv_2020/papers_ECCV/papers/123460409.pdf).
>
> Hence, given a new dataset, our algorithm requires two pieces of information: bounding boxes (GT or predicted) and at least one localized object.
> Then, our algorithm works iteratively, starting from prior knowledge, i.e., at least one localized anchor/target, to localize the rest of the mentioned anchors in the utterance given the previously localized ones without the need for any human efforts.
>
> To show the influence of this module, we report the results for two variants for each baseline, one using our pseudo-noisy labels (Algorithm 1), which is human-free, and the second one using the anchors GT annotations from “PhraseRefer.”
> Significance performance gain can be shown for the MVT case in Table 4, where using our pseudo labels, we outperform “PhraseRefer” and “ScanEnts,” which utilize GT anchors annotations.
> Furthermore, using their manual annotations instead of our pseudo labels, we achieve SOTA results and outperform them significantly.
> This emphasizes the importance of our proposed approach.
>
> > 3: Considering the elaborate process of pseudo label generation, the propose frameworks seems not "data efficient" in data preparation, though it is data efficient in 3D grounding training.
>
> We believe our answers to the previous two questions should answer your concerns and alleviate any misunderstanding about how our pseudo-label generator works.
>
> If any more clarifications are needed, please let us know; we would love to elaborate more.
>
> > 4: The interpretability is limited. From the input end, GPT-3.5 is used for extract the logical order of objects given an input utterance, thus adding interpretability. But the logical relations of the objects is not explicit modeled. Instead, it is implicit learned in the Seq-2-Seq network.
>
> We argue that modeling the relations implicitly is enough, as to predict the logical order of objects correctly is a much more complex problem.
> To solve it, the model has to extract the relations correctly first.
> To probe our claim, we conducted an experiment where we added a new training objective for the language head.
> Instead of only predicting the target of the anchors from the input utterance, we add another loss on predicting the relationship between the objects as pairs and use [bipartite loss](https://arxiv.org/abs/2005.12872).
> However, as shown in the following Table, there is no gain from learning the relations explicitly.
>
> | **Method** | **Grounding Accuracy** |
> |----------|------------------------|
> | **MVT**                    | 55.1 |
> | **MVT+CoT**                | **64.4** |
> | **MVT+CoT+Realtions Objective**                | **64.2** |

---

### Official Review · Reviewer_6KUg · 2023-10-28

**Soundness:** 3 good
**Presentation:** 3 good
**Contribution:** 3 good
**Rating:** 6
**Confidence:** 5

**Summary:**

This paper proposed to model the referring task into a seq2seq task. The authors introduce two new modules: “target anchors head” and “pathway head”.  Basically, the main idea behind is to force the model not only to predict the target object but also pay attention to the rest object mentioned in the utterance. The “pathway head” forces the model to learn the logical order information that indicates how the mentioned target object is identified. However this module requires extra annotation, which they obtain from prompting the Language Language Models (LLMs). Benefit from extra clues, they improve the grounding performance on Various grounding datasets.

**Strengths:**

Strengths:
1. This paper introduces not only just predict the target object, but also to force the model to predict all mentioned objects (anchors) to best leverage all available clues. Which provides more information for models to use.
2. The proposed pathway generation method leverages the ability of powerful LLMs (GPT), freeing the burden of human annotators and making the system easy to scale up.
3. The proposed pipeline not only improve the performance compare with previous SOTA, but also has higher data efficiently when training data is limited. This is valuable since 3D visual language data is not always easy to obtain.

**Weaknesses:**

Weaknesses:
1. The proposed idea which pays attention to not just the target object, has been proposed before in “PhraseRefer”. They use human annotators to obtain fine grained annotation. Which share the same underlying idea with this paper.
To me one of the major novelties of this paper is that they explicitly model the logical order while the “PhaseRefer '' uses implicit manner. I think these two works are highly related, but the author didn’t give enough introduction of this “PhraseRefer”, which makes it a bit difficult to understand the performance comparison e.g table 4. SAT with PhraseRefer/ScanEnts and yours. Besides, I believe some missing number in this table.4 e.g. MVT with PhaseRefer.
The authors should give more clear comparison with this line of works that use fine grain annotation.

2. Do all numbers in Table 4 use a GPT-generated path? If not, answer my below question:
If you have access to the GT anchor label either from “PhaseRefer” or “ScanEnts”, but you still require GPT to give you extra information “the Path”, how is the performance going in this case?


3. In Sec.3.3 Anchors parser “Then, we match the objects to their closest matching class from the ScanNet labels”, Would you explain this in more detail? Correct me if I am wrong: I assume if you assign a more fine grain label set , like ScanNet200, you are actually providing richer semantic knowledge to the model. My question is how do you assign the label, do you use the same level of semantic information with the others?

4. From the numbers of MVT in Table.4 and Table.5, it seems that when MVT baseline has access to more data, the performance can improve quite a lot, So it is possible that your proposed method introduce explicit knowledge to the model while MVT can has a chain to learn that out from the data if the dataset is large enough?

**Questions:**

I have listed my question above.

---

> ### Author Response · Authors · 2023-11-21
> **Answering Points #1 and #2**
>
> We thank you for your valuable and thoughtful feedback.
> We are encouraged that you find our paper novel, effective, and reasonable and that our experiments are comprehensive with state-of-the-art results.
> Below, we will address your concern and incorporate all the feedback.
>
> > 1: The authors should give more clear comparison with this line of works that use fine grain annotation, i.e., “PhraseRefer” and “ScanEnts”.
>
> Indeed, contrasting our work with the two recent works, “PhraseRefer” and “ScanEnts,” is essential, and we incorporated this response into the discussion section to clearly position our contributions compared to these works.
>
> We acknowledge the great work proposed by “PhraseRefer” and “ScanEnts,” which paved the way for the importance of paying attention not just to the target object but also to the anchors.
> Thus, underlying approaches have some similarities in terms of demonstrating the importance of including anchors in the pipeline.
> However, there are a lot of significant differences:
> 1) Design a CoT framework for 3D visual grounding that explicitly models causal reasoning while interpreting the instruction inspired by the human perception system.
> 2) Show that we can design an efficient framework that achieves state-of-the-art results on three challenging benchmarks, i.e., Nr3D, Sr3D, and ScanRefer, without requiring human labels. Proper design of such an approach is pivotal in attaining a good performance  while circumventing the need for intensive human annotations.
> A pertinent example can be drawn from the labeling procedure employed in PhraseRefer, which demanded a cumulative workforce commitment of 3664 hours, roughly equivalent to an extensive five-month timespan.
>
> Thus, as shown in Table 4, we integrated our CoT architecture into four different baselines, i.e., LAR, SAT, MVT, and ViL.
> Our framework shows a consistent gain across different baselines.
> Most importantly, we report the results for two framework variants for each baseline, one using our pseudo-noisy labels (human-free) and the second using the anchors GT annotations from “PhraseRefer”.
> Significance performance gain can be shown for the MVT case in the Table below, where using our pseudo labels, we outperform “PhraseRefer” and “ScanEnts,” which utilize GT anchors annotations.
> Furthermore, when we use their manual annotations instead of our pseudo labels, we achieve SOTA results and outperform them significantly.
> This emphasizes the importance of our proposed CoT framework.
>
> | **Method** | **Grounding Accuracy** |
> |----------|------------------------|
> | **MVT**                    | 55.1 |
> | **MVT+ScanEnts**           | 59.3 |
> | **MVT+PhraseRefer**           | 59.0 |
> | **MVT+CoT(Pseudo labels)**                | 60.4 |
> | **MVT+CoT(GT labels)**                | **64.4** |
>
> We hope this comparison clarifies the differences and poses = work clearly within the literature, specifically “PhraseRefer” and “ScanEnts”.
>
> > 2: I believe some missing number in this TableTable.4 e.g. MVT with PhaseRefer.
>
> Thank you so much for pointing out this point.
> We missed this number as it was added in the second revised version of the PhaseRefer paper.
> We incorporated this in the revised version in Table 4 and Table 6.

---

> ### Author Response · Authors · 2023-11-21
> **Answering Points #3, #4 and #5**
>
> > 3: Do all numbers in Table 4 use a GPT-generated path? If not, then, If you have access to the GT anchor label either from “PhaseRefer” or “ScanEnts”, but you still require GPT to give you extra information “the Path”, how is the performance going in this case?
>
> We do not use the GPT-generated path during inference or test time.
>
> The GPT-generated path is only used as a teacher label for training a simple MLP that predicts the path during the inference time.
> Therefore, GPT is not involved in the system during inference, as we train a simple MLP layer, i.e., the Pathway-Head in Figure 3, to predict the logical order.
> For instance, when the system is deployed on a robot, it will be efficient as our simple MLP Pathway-Head is utilized to generate a reasonable order based on the input description.
>
> Accordingly, GPT could be seen as a design choice to obviate the need for human annotations during training.
>
> In the Table below, the Pseudo and GT labels are for the anchors' localization, not the path.
> For the path, we always use our MLP-predicted path.
> However, the accuracy will not be affected if we replace it with the GPT-generated path.
> As shown in the Table below, we achieve 64.4% and 64.5% using our predicted and GPT-generated paths, respectively.
> This is an intuitive result, as when we assess the performance of the Pathway Head separately, we get 98% accuracy w.r.t the GPT-generated path.
> Therefore, replacing the Pathway Head generated path by the GPT ones is expected not to impact the final performance.
>
> | **Method** | **Grounding Accuracy** |
> |----------|------------------------|
> | **MVT**                    | 55.1 |
> | **MVT+ScanEnts**           | 59.3 |
> | **MVT+PhraseRefer**           | 59.0 |
> | **MVT+CoT(Pseudo labels)**                | 60.4 |
> | **MVT+CoT(GT labels)**                | 64.4 |
> | **MVT+CoT(GT labels+GPT Path)**                | 64.5 |
>
> This experiment is added in the revised version in the Appendix section.
>
>
> > 4: In Sec.3.3 Anchors parser “Then, we match the objects to their closest matching class from the ScanNet labels”, Would you explain this in more detail?
>
> Due to the free-from nature of Nr3D, the anchors mentioned in the GT descriptions sometimes do not precisely match the ScanNet class labels.
> For instance, the GT description is “The plant at the far right-hand side of the bookcase tucked in the furthest corner of the desk.” However, there is no “bookcase” class in ScanNet.
>
> Therefore, we need to match it to the nearest ScanNet class labels, which in this case will be “bookshelf.”
> Thus, we first check whether the extracted anchor from the text matches one of the ScanNet class labels.
> If not, then we encode the anchor words and the complete list of the ScanNet class labels.
> Then, we measure the cosine similarity on the feature space to get the closest one.
> Considering that the anchor's name may contain several words, e.g., office chair or file cabinet, we can not use BERT encodings.
> Instead, we leverage its extended version, capable of encoding sentence level instead of word level, termed [SBERT](https://arxiv.org/abs/1908.10084).
>
> > 5: My question is how do you assign the label, do you use the same level of semantic information with the others?
>
> Yes, we use the same semantic information level as the other work for a fair comparison.
> However, it is straightforward to use ScanNet200 instead for richer semantics, but then we have to retrain all the existing work on ScanNet200 for fair comparison.
> That's why, to avoid this and for a fair comparison, we adhere to the well-established setup by other related work.

---

> ### Author Response · Authors · 2023-11-21
> **Answering Point # 6**
>
> > 6: From the numbers of MVT in Table.4 and Table.5, it seems that when MVT baseline has access to more data, the performance can improve quite a lot.
> So it is possible that your proposed method introduce explicit knowledge to the model while MVT can has a chain to learn that out from the data if the dataset is large enough?
>
> To study the effect of adding more data on the performance of our method compared to MVT, we will explore three case studies:
>
> 1) **Nr3D+Sr3D results**
> From Table 4 and Table 5, when we use more data by training on Sr3D and Nr3D jointly instead of Nr3D alone, the MVT performance increased by 3.4% (from 55.1 to 58.5).
> On the other hand, our CoT approach performance increases by 2% (from 60.4 to 62.5).
> When more data is used, this reduction in the performance gain could be considered a minor impact because the gap is only reduced by 1% despite increasing the training data 3 times.
> Wherease the Nr3D dataset consists of 41.5K samples, while the Sr3D dataset contains 83.5K samples.
>
> 2) **ScanRefer results**
> As shown in Table 6, our method introduces a consistent gain.
> For instance, using 70% of the data, we outperform MVT by 8%, where the grounding accuracies are 62.5% and 54.5% for our method and MVT, respectively.
> Moreover, when more data is used, where 100% of the data is used instead of 70%, we still outperform MVT by 7%.
>
> 3) **Nr3D and Sr3D results**
> As shown in Figure 2, our CoT framework introduces a consistent gain across different amounts of training data (10% - 100%) on four different baselines.
>
> In conclusion, the aforementioned case studies show that adding more data enhances both MVT and our method equally, which implies that our CoT approach introduces a significant gain that can not be replaced by just scaling the data.

---

> ### Comment · Reviewer_6KUg · 2023-11-22
> **Reviewer response**
>
> Thanks for your response and effort. All my concerns are addressed.
>
> I would like to increase my rating from a 6 (borderline accept) to a 7 (weak accept), but unfortunately, the system does not provide this option this year. As the paper does not meet my criteria for an 8, I will maintain my rating at 6. However, I want to emphasize that my preferred rating would be a **7**.

---

### Official Review · Reviewer_ywux · 2023-11-01

**Soundness:** 3 good
**Presentation:** 3 good
**Contribution:** 3 good
**Rating:** 6
**Confidence:** 4

**Summary:**

This paper presents CoT3DRef, a novel and interpretable 3D visual grounding framework that aims to mimic the human perception system. The authors formulate the 3D visual grounding problem as a sequence-to-sequence task, predicting a chain of anchors leading up to the final target, which is a significant deviation from existing methods that directly localize the referred object. This approach not only improves the overall performance but also enhances interpretability, helping to identify and address failure cases.

**Strengths:**

1. **Technical Novelty.** The chain-of-thoughts (CoT) approach is an innovative way of addressing the 3D visual grounding problem, providing a sequence of interpretable intermediate steps that lead to the final prediction. This approach is aligned with how humans might approach the task, thereby making the model’s predictions more understandable and transparent.

2. **Data Efficiency.** The proposed framework demonstrates significant data efficiency, especially highlighted by its performance on the Sr3D dataset where it matches state-of-the-art (SOTA) performance with just 10% of the training data. This is a crucial advantage, particularly in domains where acquiring labeled data is expensive and time-consuming.

3. **Applicability.** The authors show that CoT3DRef can be easily integrated into various existing architectures, demonstrating its versatility and applicability. This is substantiated by the comprehensive experiments conducted across different benchmarks and architectures, consistently showing performance gains.

4. **Comprehensive Experiments.** The paper includes extensive experiments and ablation studies, providing a thorough evaluation of the proposed method. The results on Nr3D, Sr3D, and ScanRefer benchmarks are impressive, showcasing the framework's effectiveness and robustness.

**Weaknesses:**

1. **Detailed Failure Case Analysis.** While the paper mentions the identification of failure cases as a benefit of the interpretability, it does not provide a detailed analysis or examples of these cases. Including such an analysis could provide valuable insights into the limitations of the model and areas for future improvement.

2. **Broader Impact and Ethical Considerations.** The paper could discuss the broader impacts and potential ethical considerations of deploying such a system, especially in the highlighted application areas like autonomous driving and robotics.

**Questions:**

Can you provide a detailed analysis or examples of the identification of failure cases as a benefit of the interpretability?

---

> ### Author Response · Authors · 2023-11-21
>
> We thank you for your valuable and thoughtful feedback.
> We are encouraged that you find our paper novel, effective, and reasonable and that our experiments are comprehensive with state-of-the-art results.
> Below, we will address your concern and incorporate all the feedback.
>
> > 1: **Detailed Failure Case Analysis**. Can you provide a detailed analysis or examples of the identification of failure cases as a benefit of the interpretability?
>
> To demonstrate our interpretability abilities, we visualize attention maps as shown in Figure 5 in the Appendix.
> In Figure 5, the input description is “the chair to the far left hand side of the taller desk with the computer monitor on it.”
> Accordingly, there are two anchors mentioned in the description, i.e., desk and monitor, and the target is the chair. The correct chair should be number two, however, the model predicts number four.
> By visualizing the attention maps, on the left part of Figure 5, we can identify the main cause of the wrong prediction, whereas, the first anchor localize a wrong desk (desk \#3) instead of desk \#2.
> Therefore, the rest of the chain, i.e., the monitor and the chair are localized wrongly.
> This example also shows the ambiguity in Nr3D, where it is hard to say which desk is the taller, desk \#2 or desk \#3.
>
> > 2: Broader Impact and Ethical Considerations
>
> The paramount consideration of social impact has significantly shaped the development of our proposed CoT 3D visual grounding architecture.
> In Section 5, we had a discussion on the social impact of the technologies that may benefit from our work.
> Due to the importance of the topic as pointed by the reviewer, we have extended it in the Appendix to cover the highlighted applications like autonomous driving and robotics.
>
> 3D visual grounding holds profound implications across diverse domains and applications, spanning from integrating outdoor systems, e.g., autonomous driving and indoor navigation systems.
>
> **Outdoor navigation systems:**
> 3D visual grounding empowers the agents to execute a broad spectrum of tasks, facilitating assistive systems catering to the needs of elderly individuals and those with disabilities—such as the visually impaired.
> For instance, in the context of visually impaired individuals, the technology's ability to ground objects within their 3D environment from language descriptions is pivotal in navigation tasks such as determining the nearest exit or chair.
>
> **Autonomous driving:**
> In the realm of autonomous driving, a 3D visual grounding system takes on a critical role in translating navigational instructions into actionable responses within the dynamic visual landscape.
> This technology enables seamless communication between the language input, such as navigation commands, and the 3D world the autonomous vehicle operates in.
> For instance, when given a directive like "merge into the right lane at the upcoming intersection where the three pedestrians are walking," the 3D visual grounding system interprets this command by analyzing the spatial layout, recognizing lanes and pedestrians, and identifying the appropriate merging point. By bridging the gap between language instructions and the complex visual cues inherent in driving scenarios, this system enhances the vehicle's ability to accurately interpret and execute instructions, contributing to safer and more efficient autonomous driving experiences.
>
> **Data-efficiency importance for autonomous driving:**
> Introducing a data-efficient solution for the 3D visual grounding task holds immense importance in the context of autonomous driving, particularly considering the challenges associated with manually labeling vast amounts of 3D point cloud data from lidar sensors.
> The conventional approach of manually annotating such data is labor-intensive and economically burdensome.
> By proposing a data-efficient solution, our system addresses a critical bottleneck in developing autonomous driving technology.
> This allows the autonomous vehicle to learn and generalize from minimal labeled data, significantly reducing the dependency on large, expensive datasets.
> The capacity to make accurate 3D visual groundings with limited labeled information not only streamlines the training process but also makes deploying autonomous vehicles more scalable and cost-effective.
>
> **Federated learning for indoor navigation systems:**
> Aligned with this imperative, the inherent data-efficient nature of our approach positions it as an ideal candidate for federated learning schemes, which is perfectly aligned with the indoor setup when privacy is required, e.g., the robot is operating in a personalized environment such as the client home.
> This becomes particularly relevant when a robot is introduced into a novel indoor environment and necessitates rapid learning from minimal interactions with its new user without sending the private data of the new environment to the server; thus, learning from limited data is essential.

---

### Official Review · Reviewer_hPTL · 2023-11-01

**Soundness:** 3 good
**Presentation:** 3 good
**Contribution:** 3 good
**Rating:** 6
**Confidence:** 4

**Summary:**

This submission studies 3D object referring. A new architecture is proposed, which predicts several objects sequentially instead of a single object at last. To generate a dataset that makes this new task setting possbile, the authors exploit grammar parser, GPT in-context logical ordering and rule-based box matching. The new architecture boils down to a decoder that matches several different encoder. Experiments show that the method improves several baselines and notably work quite well under a data efficient setting.

**Strengths:**

+ A new decoder head for sequential object grounding in a scene. This is a new paradigm that may inspired related fields.
+ A data curation effort that makes sense to me. This makes the CoT training possible.
+ Quantitative results showing positive margins on four baselines and notably good performance in a data-efficient setting.

**Weaknesses:**

I like the paper and don't see major weaknesses. Why I am not rating higher is the fact that the new thing comes from foundation models, which is in fact a system-level contribution instead of a principled contribution. But I feel it fine and timely to accept this in ICLR 2024.

**Questions:**

None

---

> ### Author Response · Authors · 2023-11-21
> **GPT is used only during preparing Pseudo GT labels to train the logical order module**
>
> We thank you for your valuable and thoughtful feedback.
> We are encouraged that you like our paper and find our approach effective and reasonable and our experiments comprehensive with state-of-the-art results.
> Below, we will address your concern and incorporate all the feedback.
>
> > 1: I like the paper and don't see major weaknesses. Why I am not rating higher is the fact that the new thing comes from foundation models, which is in fact a system-level contribution instead of a principled contribution.
>
> Our main contribution in this paper is demonstrating that it is possible to design a CoT framework for 3D visual grounding efficiently without human labels. It also shows that it can be integrated with multiple existing methods, showing significant gains, especially when data is limited.
>
>
> Therefore, the foundation model, i.e., GPT, is used only for the logical order ground-truth preparation, used during the training.
> Specifically, it is not involved in the system during inference, as we train a simple MLP layer, i.e., the Pathway-Head in Figure 3, to predict the logical order.
> Thus, for instance, when the system is deployed on a robot, it will be efficient as a simple MLP is utilized to generate a reasonable order based on the input description.
>
> Accordingly, the foundation model could be seen as a design choice to obviate the need for human annotations, to guide the training process.
> No foundation models are loaded or used at test time, making the model inference-time efficient.

---

> > ### Comment · Reviewer_hPTL · 2023-11-22
> > **Feedback**
> >
> > Yes, I understand that GPT is only used during training but not testing.

---

### Author Response · Authors · 2023-11-21
**Summarizing the Key Points in our Response**

We thank the reviewers for their valuable and thoughtful feedback. We are encouraged that reviewers find our paper novel, effective, and reasonable and that our experiments are comprehensive with state-of-the-art results.
We will address your concerns and questions individually and incorporate all the feedback.

We include the following updates in our response:

* A revised version of our paper is attached and the updates are marked in blue.

* Identification of failure cases as a benefit of the interpretability.

* Broader Impact and Ethical Considerations.

* Add an ablation study for the GPT-generated path influence on the overall grounding accuracy.

* Add more clarifications for the localization algorithm, Algorithm 1.

* Answer the reviewer's questions and clarify some points, such as our contributions and the limited role of GPT while only preparing Pseudo GT labels for the logical order module.

During the remaining discussion period, please let us know if any concerns or further clarifications are needed, and we will do our best to respond immediately.

Thank you once again for your time and valuable feedback.

---

### Meta-Review · Area_Chair_UkJD · 2023-12-09

**Metareview:**

Paper propose an approach for 3D referring expression grounding. Unlike current approaches that leverage text encoder or vision-language encoder and output the final 3D box, this approach relies on a sequence of object localizations to contextualize the final answer (similar to Chain-of-Thought in language). All reviewers rate the paper as Marginally Above Acceptance (with [6KUg] mentioning in comments that he/she would like to raise the score).

All reviewers agree that chain of thought is an interesting and innovative way to address 3D visual grounding. Some questions and comments were raised by reviewers but appear to be largely addressed by the rebuttal. There is some concern from [hPTL] that contributions are "system-level" instead of more "principled". However, at the same time [hPTL] argues for the acceptance and timeliness of the work.

AC has read the reviews, rebuttal, discussion and looked at the paper itself. AC agrees with reviewers that proposed approach is interesting, timely and could potentially be broadly beneficial (having impact beyond 3D grounding). AC believes that contributions are sufficient and the paper should be Accepted.

**Justification For Why Not Higher Score:**

Personally, I actually would not mind this being a Spotlight (I quite like the idea and it appears relatively general), but given the Reviewer scores didn't feel that was necessarily justifiable.

**Justification For Why Not Lower Score:**

The work is interesting and novel. Idea to do Visual Chain of Thought is interesting and quite innovative. I think it can be useful more broadly.

---

### Decision · Program_Chairs · 2024-01-16

Accept (poster)